# Decoupling Exploration and Exploitation for Meta-Reinforcement Learning without Sacrifices

## Abstract

The goal of meta-reinforcement learning (meta-RL) is to build agents that can quickly learn new tasks by leveraging prior experience on related tasks. Learning a new task often requires both exploring to gather task-relevant information and exploiting this information to solve the task. In principle, optimal exploration and exploitation can be learned end-to-end by simply maximizing task performance. However, such meta-RL approaches struggle with local optima due to a chicken-and-egg problem: learning to explore requires good exploitation to gauge the exploration's utility, but learning to exploit requires information gathered via exploration. Optimizing separate objectives for exploration and exploitation can avoid this problem, but prior meta-RL exploration objectives yield suboptimal policies that gather information irrelevant to the task. We alleviate both concerns by constructing an exploitation objective that automatically identifies task-relevant information and an exploration objective to recover only this information. This avoids local optima in end-to-end training, without sacrificing optimal exploration. Empirically, DREAM substantially outperforms existing approaches on complex meta-RL problems, such as sparse-reward 3D visual navigation.[1]

## 1 Introduction

A general-purpose agent should be able to perform multiple related tasks across multiple related environments. Our goal is to develop agents that can perform a variety of tasks in novel environments, based on previous experience and only a small amount of experience in the new environment. For example, we may want a robot to cook a meal (a new task) in a new kitchen (the environment) after it has learned to cook other meals in other kitchens. To adapt to a new kitchen, the robot must both explore to find the ingredients, and use this information to cook. Existing meta-reinforcement learning (meta-RL) methods can adapt to new tasks and environments, but, as we identify in this work, struggle when adaptation requires complex exploration.

In the meta-RL setting, the agent is presented with a set of meta-training problems, each in an environment (e.g., a kitchen) with some task (e.g., make pizza); at meta-test time, the agent is given a new, but related environment and task. It is allowed to gather information in a few initial (exploration) episodes, and its goal is to then maximize returns on all subsequent (exploitation) episodes, using this information. A common meta-RL approach is to learn to explore and exploit *end-to-end* by training a policy and updating exploration behavior based on how well the policy later exploits using the information discovered from exploration (Duan et al., 2016; Wang et al., 2016a; Stadie et al., 2018; Zintgraf et al., 2019; Humplik et al., 2019). With enough model capacity, such approaches can express optimal exploration and exploitation, but they create a chicken-and-egg problem that leads to bad local optima and poor sample efficiency: Learning to explore requires good exploitation to gauge the exploration's utility, but learning to exploit requires information gathered via exploration; therefore, with only final performance as signal, one cannot be learned without already having learned the other. For example, a robot chef is only incentivized to explore and find the ingredients if it already knows how to cook, but the robot can only learn to cook if it can already find the ingredients by exploration.

To avoid the chicken-and-egg problem, we propose to optimize separate objectives for exploration and exploitation by leveraging the *problem ID*—an easy-to-provide unique one-hot for each training meta-

---

[1]Project web page: `https://anonymouspapersubmission.github.io/dream/`

training task and environment. Such a problem ID can be realistically available in real-world meta-RL tasks: e.g., in a robot chef factory, each training kitchen (problem) can be easily assigned a unique ID, and in a recommendation system that provides tailored recommendations to each user, each user (problem) is typically identified by a unique username. Some prior works (Humplik et al., 2019; Kamienny et al., 2020) also use these problem IDs, but not in a way that avoids the chicken-and-egg problem. Others (Rakelly et al., 2019; Zhou et al., 2019b; Gupta et al., 2018; Gurumurthy et al., 2019; Zhang et al., 2020) also optimize separate objectives, but their exploration objectives learn suboptimal policies that gather task-irrelevant information (e.g., the color of the walls). Instead, we propose an exploitation objective that automatically identifies task-relevant information, and an exploration objective to recover only this information. We learn an exploitation policy without the need for exploration, by conditioning on a learned representation of the problem ID, which provides task-relevant information (e.g., by memorizing the locations of the ingredients for each ID / kitchen). We also apply an information bottleneck to this representation to encourage discarding of any information not required by the exploitation policy (i.e., task-irrelevant information). Then, we learn an exploration policy to only discover task-relevant information by training it to produce trajectories containing the same information as the learned ID representation (Section 4). Crucially, unlike prior work, we prove that our separate objectives are *consistent*: optimizing them yields optimal exploration and exploitation, assuming expressive-enough policy classes and enough meta-training data (Section 5.1).

Overall, we present two core contributions: (i) we articulate and formalize a chicken-and-egg coupling problem between optimizing exploration and exploitation in meta-RL (Section 4.1); and (ii) we overcome this with a consistent decoupled approach, called DREAM: **D**ecoupling explo**R**ation and **E**xploit**A**tion in **M**eta-RL (Section 4.2). Theoretically, in a simple tabular example, we show that addressing the coupling problem with DREAM provably improves sample complexity over existing end-to-end approaches by a factor exponential in the horizon (Section 5). Empirically, we stress test DREAM's ability to learn sophisticated exploration strategies on 3 challenging, didactic benchmarks and a sparse-reward 3D visual navigation benchmark. On these, DREAM learns to optimally explore and exploit, achieving 90% higher returns than existing state-of-the-art approaches (PEARL, E-RL$^2$, IMPORT, VARIBAD), which struggle to learn an effective exploration strategy (Section 6).

## 2 RELATED WORK

We draw on a long line of work on learning to adapt to related tasks (Schmidhuber, 1987; Thrun & Pratt, 2012; Naik & Mammone, 1992; Bengio et al., 1991; 1992; Hochreiter et al., 2001; Andrychowicz et al., 2016; Santoro et al., 2016). Many meta-RL works focus on adapting efficiently to a new task from few samples without optimizing the sample collection process, via updating the policy parameters (Finn et al., 2017; Agarwal et al., 2019; Yang et al., 2019; Houthooft et al., 2018; Mendonca et al., 2019), learning a model (Nagabandi et al., 2018; Sæmundsson et al., 2018; Hiraoka et al., 2020), multi-task learning (Fakoor et al., 2019), or leveraging demonstrations (Zhou et al., 2019a). In contrast, we focus on problems where targeted exploration is critical for few-shot adaptation.

Approaches that specifically explore to obtain the most informative samples fall into two main categories: *end-to-end* and *decoupled* approaches. End-to-end approaches optimize exploration and exploitation end-to-end by updating exploration behavior from returns achieved by exploitation (Duan et al., 2016; Wang et al., 2016a; Mishra et al., 2017; Rothfuss et al., 2018; Stadie et al., 2018; Zintgraf et al., 2019; Humplik et al., 2019; Kamienny et al., 2020; Dorfman & Tamar, 2020). These approaches can represent the optimal policy (Kaelbling et al., 1998), but they struggle to escape local optima due to a chicken-and-egg problem between learning to explore and learning to exploit (Section 4.1). Several of these approaches (Humplik et al., 2019; Kamienny et al., 2020) also leverage the problem ID during meta-training, but they still learn end-to-end, so the chicken-and-egg problem remains.

Decoupled approaches instead optimize separate exploration and exploitation objectives, via, e.g., Thompson-sampling (TS) (Thompson, 1933; Rakelly et al., 2019), obtaining exploration trajectories predictive of dynamics or rewards (Zhou et al., 2019b; Gurumurthy et al., 2019; Zhang et al., 2020), or exploration noise (Gupta et al., 2018). While these works do not identify the chicken-and-egg problem, decoupled approaches coincidentally avoid it. However, existing decoupled approaches, including those (Rakelly et al., 2019; Zhang et al., 2020) that leverage the problem ID, do not learn optimal exploration: TS (Rakelly et al., 2019) explores by guessing the task and executing a policy for that task, and hence cannot represent exploration behaviors that are different from exploitation (Russo et al., 2017). Predicting the dynamics (Zhou et al., 2019b; Gurumurthy et al., 2019; Zhang et al., 2020) is inefficient when only a small subset of the dynamics are relevant to solving the task. In

Figure 1: Meta-RL setting: Given a new environment and task, the agent is allowed to first explore and gather information, and then must use this information to solve the task in subsequent exploitation episodes.

contrast, we propose a separate mutual information objective for exploration, which both avoids the chicken-and-egg problem and yields optimal exploration when optimized (Section 5). Past work (Gregor et al., 2016; Houthooft et al., 2016; Eysenbach et al., 2018; Warde-Farley et al., 2018) also optimize mutual information objectives, but not for meta-RL.

**Exploration in general RL.** The general RL setting (i.e., learning from scratch) also requires targeted exploration to gather informative samples that enables learning a policy to solve the problem. In contrast to exploration algorithms for general RL (Bellemare et al., 2016; Pathak et al., 2017; Burda et al., 2018; Leibfried et al., 2019), which must visit many novel states to find regions with high reward, exploration in meta-RL can be even more targeted by leveraging prior experience from different problems during meta-training. As a result, DREAM can learn new tasks in just *two* episodes (Section 6), while learning from scratch can require thousands or even millions of episodes.

## 3 PRELIMINARIES

**Meta-reinforcement learning.** The meta-RL setting considers a family of Markov decision processes (MDPs) $\langle \mathcal{S}, \mathcal{A}, \mathcal{R}_\mu, T_\mu \rangle$ with states $\mathcal{S}$, actions $\mathcal{A}$, rewards $\mathcal{R}_\mu$, and dynamics $T_\mu$, indexed by a one-hot *problem ID* $\mu \in \mathcal{M}$, drawn from a distribution $p(\mu)$. Colloquially, we refer to the dynamics as the *environment*, the rewards as the *task*, and the entire MDP as the *problem*. Borrowing terminology from Duan et al. (2016), meta-training and meta-testing both consist of repeatedly running *trials*. Each trial consists of sampling a problem ID $\mu \sim p(\mu)$ and running $N + 1$ episodes on the corresponding problem. Following prior evaluation settings (Finn et al., 2017; Rakelly et al., 2019; Rothfuss et al., 2018; Fakoor et al., 2019), we designate the first episode in a trial as an *exploration* episode consisting of $T$ steps for gathering information, and define the goal as maximizing the returns in the subsequent $N$ *exploitation* episodes (Figure 1). Following Rakelly et al. (2019); Humplik et al. (2019); Kamienny et al. (2020), the easy-to-provide problem ID is available for meta-training, but not meta-testing trials.

We formally express the goal in terms of an exploration policy $\pi^{\text{exp}}$ used in the exploration episode and an exploitation policy $\pi^{\text{task}}$ used in exploitation episodes, but these policies may be the same or share parameters. Rolling out $\pi^{\text{exp}}$ in the exploration episode produces an exploration trajectory $\tau^{\text{exp}} = (s_0, a_0, r_0, \ldots, s_T)$, which contains information discovered via exploration. The exploitation policy $\pi^{\text{task}}$ may then condition on $\tau^{\text{exp}}$ and optionally, its history across all exploitation episodes in a trial, to maximize exploitation episode returns. The goal is therefore to maximize:

$$\mathcal{J}(\pi^{\text{exp}}, \pi^{\text{task}}) = \mathbb{E}_{\mu \sim p(\mu), \tau^{\text{exp}} \sim \pi^{\text{exp}}} \left[ V^{\text{task}}(\tau^{\text{exp}}; \mu) \right], \tag{1}$$

where $V^{\text{task}}(\tau^{\text{exp}}; \mu)$ is the expected returns of $\pi^{\text{task}}$ conditioned on $\tau^{\text{exp}}$, summed over the $N$ exploitation episodes in a trial with problem ID $\mu$.

**End-to-end meta-RL.** A common meta-RL approach (Wang et al., 2016a; Duan et al., 2016; Rothfuss et al., 2018; Zintgraf et al., 2019; Kamienny et al., 2020; Humplik et al., 2019) is to learn to explore and exploit *end-to-end* by directly optimizing $\mathcal{J}$ in (1), updating both from rewards achieved during exploitation. These approaches typically learn a single recurrent policy $\pi(a_t \mid s_t, \tau_{:t})$ for both exploration and exploitation (i.e., $\pi^{\text{task}} = \pi^{\text{exp}} = \pi$), which takes action $a_t$ given state $s_t$ and history of experiences spanning all episodes in a trial $\tau_{:t} = (s_0, a_0, r_0, \ldots, s_{t-1}, a_{t-1}, r_{t-1})$. Intuitively, this policy is learned by rolling out a trial, producing an exploration trajectory $\tau^{\text{exp}}$ and, conditioned on $\tau^{\text{exp}}$ and the exploitation experiences so far, yielding some exploitation episode returns. Then, credit is assigned to both exploration (producing $\tau^{\text{exp}}$) and exploitation by backpropagating the exploitation returns through the recurrent policy. Critically, estimates of the expected exploitation returns in (1) (e.g., from a single roll-out or value-function approximation) form the learning signal for exploration. Directly optimizing the objective $\mathcal{J}$ this way can learn optimal exploration and exploitation strategies, but optimization is challenging, which we show in Section 4.1.

Figure 2: (a) Coupling between the exploration policy $\pi^{\text{exp}}$ and exploitation policy $\pi^{\text{task}}$. These policies are illustrated separately for clarity, but may be a single policy. Since the two policies depend on each other (for gradient signal and the $\tau^{\text{exp}}$ distribution), it is challenging to learn one when the other policy has not learned. (b) DREAM: $\pi^{\text{exp}}$ and $\pi^{\text{task}}$ are learned from decoupled objectives by leveraging a simple one-hot problem ID during meta-training. At meta-test time, the exploitation policy conditions on the exploration trajectory as before.

## 4 DECOUPLING EXPLORATION AND EXPLOITATION

### 4.1 THE PROBLEM WITH COUPLING EXPLORATION AND EXPLOITATION

We begin by showing that end-to-end optimization struggle with local optima due to a chicken-and-egg problem. Figure 2a illustrates this. Learning $\pi^{\text{exp}}$ relies on gradients passed through $\pi^{\text{task}}$. If $\pi^{\text{task}}$ cannot effectively solve the task, then these gradients will be uninformative. However, to learn to efficiently solve the task, $\pi^{\text{task}}$ needs good exploration data (trajectories $\tau^{\text{exp}}$) from a good exploration policy $\pi^{\text{exp}}$. This results in bad local optima as follows: if our current (suboptimal) $\pi^{\text{task}}$ obtains low rewards with a good informative trajectory $\tau^{\text{exp}}_{\text{good}}$, the low reward would cause $\pi^{\text{exp}}$ to learn to *not* generate $\tau^{\text{exp}}_{\text{good}}$. This causes $\pi^{\text{exp}}$ to instead generate trajectories $\tau^{\text{exp}}_{\text{bad}}$ that lack information required to obtain high reward, further preventing the exploitation policy $\pi^{\text{task}}$ from learning. Typically, early in training, both $\pi^{\text{exp}}$ and $\pi^{\text{task}}$ are suboptimal and hence will likely reach this local optimum. In Section 5.2, we illustrate how this local optimum can cause sample inefficiency in a simple example.

### 4.2 DREAM: DECOUPLING EXPLORATION AND EXPLOITATION IN META-LEARNING

While we can sidestep the local optima of end-to-end training by optimizing separate objectives for exploration and exploitation, the challenge is to construct objectives that yield the same optimal solution as the end-to-end approach. We now discuss how we can use the easy-to-provide problem IDs during meta-training to do so. A good exploration objective should encourage discovering task-relevant distinguishing attributes of the problem (e.g., ingredient locations), and ignoring task-irrelevant attributes (e.g., wall color). To create this objective, the key idea behind DREAM is to *learn* to extract only the task-relevant information from the problem ID, which encodes all information about the problem. Then, DREAM's exploration objective seeks to recover this task-relevant information.

Concretely, DREAM extracts only the task-relevant information from the problem ID $\mu$ via a stochastic encoder $F_\psi(z \mid \mu)$. To learn this encoder, we train an exploitation policy $\pi^{\text{task}}$ to maximize rewards, conditioned on samples $z$ from $F_\psi(z \mid \mu)$, while simultaneously applying an information bottleneck to $z$ to discard information not needed by $\pi^{\text{task}}$ (i.e., task-irrelevant information). Then, DREAM learns an exploration policy $\pi^{\text{exp}}$ to produce trajectories with high mutual information with $z$. In this approach, the exploitation policy $\pi^{\text{task}}$ no longer relies on effective exploration from $\pi^{\text{exp}}$ to learn, and once $F_\psi(z \mid \mu)$ is learned, the exploration policy also learns independently from $\pi^{\text{task}}$, decoupling the two optimization processes. During meta-testing, when $\mu$ is unavailable, the two policies easily combine, since the trajectories generated by $\pi^{\text{exp}}$ are optimized to contain the same information as the encodings $z \sim F_\psi(z \mid \mu)$ that the exploitation policy $\pi^{\text{task}}$ trained on (overview in Figure 2b).

**Learning the problem ID encodings and exploitation policy.** We begin with learning a stochastic encoder $F_\psi(z \mid \mu)$ parametrized by $\psi$ and exploitation policy $\pi^{\text{task}}_\theta$ parametrized by $\theta$, which conditions on $z$. We learn $F_\psi$ jointly with $\pi^{\text{task}}_\theta$ by optimizing the following objective:

$$\underset{\psi,\theta}{\text{maximize}} \; \underbrace{\mathbb{E}_{\mu \sim p(\mu), z \sim F_\psi(z|\mu)}\left[ V^{\pi^{\text{task}}_\theta}(z; \mu) \right]}_{\text{Reward}} - \lambda \underbrace{I(z; \mu)}_{\text{Information bottleneck}}, \qquad (2)$$

where $V^{\pi^{\text{task}}_\theta}(z; \mu)$ is the expected return of $\pi^{\text{task}}_\theta$ on problem $\mu$ given and encoding $z$. The information bottleneck term encourages discarding any (task-irrelevant) information from $z$ that does not help maximize reward. Importantly, both terms are independent of the exploration policy $\pi^{\text{exp}}$. This objective is derived from forming the Lagrangian of a constrained optimization problem, where $\lambda^{-1}$ is the dual variable, detailed in Appendix E.

We minimize the mutual information $I(z; \mu)$ by minimizing a variational upper bound on it, $\mathbb{E}_\mu [D_{\text{KL}}(F_\psi(z \mid \mu)||r(z))]$, where $r$ is any prior and $z$ is distributed as $p_\psi(z) = \int_\mu F_\psi(z \mid \mu)p(\mu)d\mu$.

**Learning an exploration policy given problem ID encodings.** Once we've obtained an encoder $F_\psi(z \mid \mu)$ to extract only the necessary task-relevant information required to optimally solve each task, we can optimize the exploration policy $\pi^{\text{exp}}$ to produce trajectories that contain this same information by maximizing their mutual information $I(\tau^{\text{exp}}; z)$. We slightly abuse notation to use $\pi^{\text{exp}}$ to denote the probability distribution over the trajectories $\tau^{\text{exp}}$. Then, the mutual information $I(\tau^{\text{exp}}; z)$ can be efficiently maximized by maximizing a variational lower bound (Barber & Agakov, 2003) as follows:

$$I(\tau^{\text{exp}}; z) = H(z) - H(z \mid \tau^{\text{exp}}) \geq H(z) + \mathbb{E}_{\mu, z \sim F_\psi, \tau^{\text{exp}} \sim \pi^{\text{exp}}} [\log q_\omega(z \mid \tau^{\text{exp}})] \qquad (3)$$

$$= H(z) + \mathbb{E}_{\mu, z \sim F_\psi}[\log q_\omega(z)] + \mathbb{E}_{\mu, z \sim F_\psi, \tau^{\text{exp}} \sim \pi^{\text{exp}}} \left[ \sum_{t=1}^T \log q_\omega(z \mid \tau^{\text{exp}}_{:t}) - \log q_\omega(z \mid \tau^{\text{exp}}_{:t-1}) \right],$$

where $q_\omega$ is any distribution parametrized by $\omega$. We maximize the above expression over $\omega$ to learn $q_\omega$ that approximates the true conditional distribution $p(z \mid \tau^{\text{exp}})$, which makes this bound tight. In addition, we do not have access to the problem $\mu$ at test time and hence cannot sample from $F_\psi(z \mid \mu)$. Therefore, $q_\omega$ serves as a decoder that generates the encoding $z$ from the exploration trajectory $\tau^{\text{exp}}$.

Recall, our goal is to maximize (3) w.r.t., trajectories $\tau^{\text{exp}}$ from the exploration policy $\pi^{\text{exp}}$. Only the third term depends on $\tau^{\text{exp}}$, so we train $\pi^{\text{exp}}$ on rewards set to be this third term (information gain):

$$r_t^{\text{exp}}(a_t, s_{t+1}, \tau^{\text{exp}}_{t-1}; \mu) = \mathbb{E}_{z \sim F_\psi(z|\mu)} \left[ \log q_\omega(z \mid [s_{t+1}; a_t; \tau^{\text{exp}}_{:t-1}]) - \log q_\omega(z \mid \tau^{\text{exp}}_{:t-1}) \right] - c. \qquad (4)$$

Intuitively, the exploration reward for taking action $a_t$ and transitioning to state $s_{t+1}$ is high if this transition encodes more information about the problem (and hence the encoding $z \sim F_\psi(z \mid \mu)$) than was already present in the trajectory $\tau^{\text{exp}}_{:t-1} = (s_0, a_0, r_0, \ldots, s_{t-2}, a_{t-2}, r_{t-2})$. We also include a small penalty $c$ to encourage exploring efficiently in as few timesteps as possible. This reward is attractive because (i) it is independent from the exploitation policy and hence avoids the local optima described in Section 4.1, and (ii) it is dense, so it helps with credit assignment. It is also non-Markov, since it depends on $\tau^{\text{exp}}$, so we maximize it with a recurrent $\pi_\phi^{\text{exp}}(a_t \mid s_t, \tau^{\text{exp}}_{:t})$, parametrized by $\phi$.

## 4.3 A PRACTICAL IMPLEMENTATION OF DREAM

Altogether, DREAM learns four separate neural network components, which we detail below.

1. Encoder $F_\psi(z \mid \mu)$: For simplicity, we parametrize the stochastic encoder by learning a deterministic encoding $f_\psi(\mu)$ and apply Gaussian noise, i.e., $F_\psi(z \mid \mu) = \mathcal{N}(f_\psi(\mu), \rho^2 I)$. We choose a convenient prior $r(z)$ to be a unit Gaussian with same variance $\rho^2 I$, which makes the information bottleneck take the form of simple $\ell_2$-regularization $\|f_\psi(\mu)\|_2^2$.

2. Decoder $q_\omega(z \mid \tau^{\text{exp}})$: Similarly, we parametrize the decoder $q_\omega(z \mid \tau^{\text{exp}})$ as a Gaussian centered around a deterministic encoding $g_\omega(\tau^{\text{exp}})$ with variance $\rho^2 I$. Then, $q_\omega$ maximizes $\mathbb{E}_{\mu, z \sim F_\psi(z|\mu)} \left[ \|z - g_\omega(\tau^{\text{exp}})\|_2^2 \right]$ w.r.t., $\omega$ (Equation 3), and the exploration rewards take the form $r^{\text{exp}}(a, s', \tau^{\text{exp}}; \mu) = \|f_\psi(\mu) - g_\omega([\tau^{\text{exp}}; a; s'])\|_2^2 - \|f_\psi(\mu) - g_\omega([\tau^{\text{exp}}])\|_2^2 - c$ (Equation 4).

3. Exploitation policy $\pi_\theta^{\text{task}}$ and 4. Exploration policy $\pi_\phi^{\text{exp}}$: We learn both policies with double deep Q-learning (van Hasselt et al., 2016), treating $(s, z)$ as the state for $\pi_\theta^{\text{task}}$.

For convenience, we jointly learn all components in an EM-like fashion, where in the exploration episode, we assume $f_\psi$ and $\pi_\theta^{\text{task}}$ are fixed. Appendix A includes all details and a summary (Algorithm 1).

Overall, DREAM avoids the chicken-and-egg problem in Section 4.1. DREAM learns the exploitation policy and encoder completely independently from exploration. This enables the encoder to learn quickly, and once it has learned, it (together with the decoder) forms a learning signal for exploration separate from the expected exploitation returns, which improves sample efficiency (Section 5.2).

## 5 ANALYSIS OF DREAM

### 5.1 THEORETICAL CONSISTENCY OF THE DREAM OBJECTIVE

A key property of DREAM is that it is *consistent*: maximizing our decoupled objective also maximizes expected returns (Equation 1). This contrasts prior decoupled approaches (Zhou et al., 2019b; Rakelly

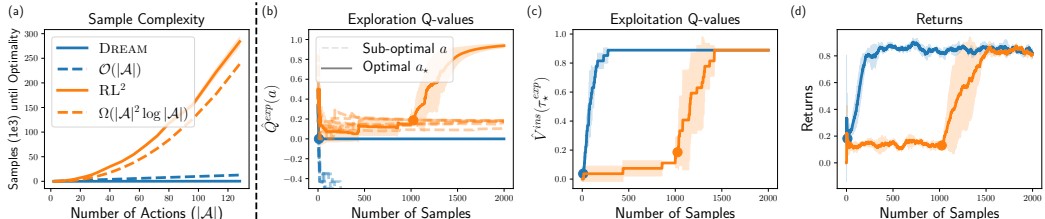

Figure 3: (a) Sample complexity of learning the optimal exploration policy as the action space $|\mathcal{A}|$ grows (1000 seeds). (b) Exploration Q-values $\hat{Q}^{\mathrm{exp}}(a)$. The policy $\arg\max_a \hat{Q}^{\mathrm{exp}}(a)$ is optimal after the dot. (c) Exploitation values given optimal trajectory $\hat{V}^{\mathrm{task}}(\tau_\star^{\mathrm{exp}})$. (d) Returns achieved on a tabular MDP with $|\mathcal{A}| = 8$ (3 seeds).

et al., 2019; Gupta et al., 2018; Gurumurthy et al., 2019; Zhang et al., 2020), which also decouple exploration from exploitation, but do not recover the optimal policy even with infinite data. Formally,

**Proposition 1.** *Assume* $\langle \mathcal{S}, \mathcal{A}, \mathcal{R}_\mu, \mathcal{T}_\mu \rangle$ *is ergodic for all problems* $\mu \in \mathcal{M}$. *Let* $V^*(\mu)$ *be the maximum expected returns achievable by any exploitation policy with access to the problem ID* $\mu$, *i.e., with complete information. Let* $\pi_\star^{task}, \pi_\star^{exp}, F_\star$ *and* $q_\star(z \mid \tau^{exp})$ *be the optimizers of the* DREAM *objective. Then, if the function classes* DREAM *optimizes over are well-specified, there exists a finite* $T$ *such that if the length of the exploration episode is at least* $T$,

$$\mathbb{E}_{\mu\sim p(\mu),\tau^{exp}\sim\pi_\star^{exp},z\sim q_\star(z|\tau^{exp})}\left[V^{\pi_\star^{task}}(z;\mu)\right] = \mathbb{E}_{\mu\sim p(\mu)}\left[V^*(\mu)\right].$$

Optimizing DREAM's objective achieves the maximal returns $V^*(\mu)$ even without access to $\mu$ during meta-testing (proof in Appendix C.1). We can remove the ergodicity assumption by increasing the number of exploration episodes, and DREAM performs well on non-ergodic MDPs in our experiments.

### 5.2 An Example Illustrating the Impact of Coupling on Sample Complexity

With enough capacity, end-to-end approaches can also learn the optimal policy, but can be highly sample inefficient due to the coupling problem in Section 4.1. We highlight this in a simple tabular example to remove the effects of function approximation: Each episode is a one-step bandit problem with action space $\mathcal{A}$. Taking action $a_\star$ in the exploration episode leads to a trajectory $\tau_\star^{\mathrm{exp}}$ that reveals the problem ID $\mu$; all other actions $a$ reveal no information and lead to $\tau_a^{\mathrm{exp}}$. The ID $\mu$ identifies a unique action that receives reward 1 during exploitation; all other actions get reward 0. Therefore, taking $a_\star$ during exploration is necessary and sufficient to obtain optimal reward 1. We now study the number of samples required for RL$^2$ (the canonical end-to-end approach) and DREAM to learn the optimal exploration policy with $\epsilon$-greedy tabular Q-learning. We precisely describe a more general setup in Appendix C.2 and prove that DREAM *learns the optimal exploration policy in* $\Omega(|\mathcal{A}|^H|\mathcal{M}|)$ *times fewer samples than RL$^2$* in this simple setting with horizon $H$. Figure 3a empirically validates this result and we provide intuition below.

In the tabular analog of RL$^2$, the exploitation Q-values form targets for the exploration Q-values: $\hat{Q}^{\mathrm{exp}}(a) \leftarrow \hat{V}^{\mathrm{task}}(\tau_a^{\mathrm{exp}}) := \max_{a'} \hat{Q}^{\mathrm{task}}(\tau_a^{\mathrm{exp}}, a')$. We drop the fixed initial state from notation. This creates the local optimum in Section 4.1. Initially $\hat{V}^{\mathrm{task}}(\tau_\star^{\mathrm{exp}})$ is low, as the exploitation policy has not learned to achieve reward, even when given $\tau_\star^{\mathrm{exp}}$. This causes $\hat{Q}^{\mathrm{exp}}(a_\star)$ to be small and therefore $\arg\max_a \hat{Q}^{\mathrm{exp}}(a) \neq a_\star$ (Figure 3b), which then prevents $\hat{V}^{\mathrm{task}}(\tau_\star^{\mathrm{exp}})$ from learning (Figure 3c) as $\tau_\star^{\mathrm{exp}}$ is roughly sampled only once per $\frac{|\mathcal{A}|}{\epsilon}$ episodes. This effect is mitigated only when $\hat{Q}^{\mathrm{exp}}(a_\star)$ becomes higher than $\hat{Q}^{\mathrm{exp}}(a)$ for the other uninformative $a$'s (the dot in Figure 3b-d). Then, learning both the exploitation and exploration Q-values accelerates, but getting there takes many samples.

In DREAM, the exploration Q-values regress toward the decoder $\hat{q}$: $\hat{Q}^{\mathrm{exp}}(a) \leftarrow \log \hat{q}(\mu \mid \tau^{\mathrm{exp}}(a))$. This decoder learns much faster than $\hat{Q}^{\mathrm{task}}$, since it does not depend on the exploitation actions. Consequently, DREAM's exploration policy quickly becomes optimal (dot in Figure 3b), which enables quickly learning the exploitation Q-values and achieving high reward (Figures 3c and 3d).

In general, DREAM learns in far fewer samples than end-to-end approaches, since in end-to-end approaches like RL$^2$, exploration is learned from a quantity requiring many samples to accurately estimate (i.e., the exploitation Q-values in this case). Initially, this quantity is estimated poorly, so the signal for exploration can erroneously "down weight" good exploration behavior, which causes the

chicken-and-egg problem. In contrast, in DREAM, the exploration policy learns from the decoder, which requires far fewer samples to accurately estimate, avoiding the chicken-and-egg problem.

## 6    EXPERIMENTS

Many real-world problem distributions (e.g., cooking) require exploration (e.g., locating ingredients) that is distinct from exploitation (e.g., cooking these ingredients). Therefore, we desire benchmarks that require distinct exploration and exploitation to stress test aspects of exploration in meta-RL, such as if methods can: (i) efficiently explore, even in the presence of distractions; (ii) leverage informative objects (e.g., a map) to aid exploration; (iii) learn exploration and exploitation strategies that generalize to unseen problems; (iv) scale to challenging exploration problems with high-dimensional visual observations. Existing benchmarks (e.g., MetaWorld (Yu et al., 2019) or MuJoCo tasks like HalfCheetahVelocity (Finn et al., 2017; Rothfuss et al., 2018)) were not designed to test exploration and are unsuitable for answering these questions. These benchmarks mainly vary the rewards (e.g., the speed to run at) across problems, so naively exploring by exhaustively trying different exploitation behaviors (e.g., running at different speeds) is optimal. They further don't include visual states, distractors, or informative objects, which test if exploration is efficient. We therefore design new benchmarks meeting the above criteria, testing (i-iii) with didactic benchmarks, and (iv) with a sparse-reward 3D visual navigation benchmark, based on Kamienny et al. (2020), that combines complex exploration with high-dimensional visual inputs. To further deepen the exploration challenge, we make our benchmarks goal-conditioned. This requires exploring to discover information relevant to *any* potential goal, rather than just a single task (e.g., locating all ingredients for *any* meal vs. just the ingredients for pasta).

**Comparisons.** We compare DREAM with state-of-the-art end-to-end (E-RL$^2$ (Stadie et al., 2018), VARIBAD (Zintgraf et al., 2019), and IMPORT (Kamienny et al., 2020)) and decoupled approaches (PEARL-UB, an upper bound on the final performance of PEARL (Rakelly et al., 2019)). For PEARL-UB, we analytically compute the expected rewards achieved by optimal Thompson sampling (TS) exploration, assuming access to the optimal problem-specific policy and true posterior problem distribution. Like DREAM, IMPORT and PEARL also use the one-hot problem ID, during meta-training. We also report the optimal returns achievable with no exploration as "No exploration." Where applicable, all methods use the same architecture. The full architecture and approach details are in Appendix B.3.

We report the average returns achieved by each approach in trials with one exploration and one exploitation episode, averaged over 3 seeds with 1-standard deviation error bars (full details in Appendix B). We periodically evaluate each approach on 100 meta-testing trials, after 2000 meta-training trials. In all plots, the training timesteps includes all timesteps from both exploitation and exploration episodes in meta-training trials.

### 6.1    DIDACTIC EXPERIMENTS

We first evaluate on the grid worlds shown in Figures 4a and 4b. The state consists of the agent's $(x, y)$-position, a one-hot indicator of the object at the agent's position (none, bus, map, pot, or fridge), a one-hot indicator of the agent's inventory (none or an ingredient), and the goal. The actions are *move* up, down, left, or right; *ride bus*, which, at a bus, teleports the agent to another bus of the same color; *pick up*, which, at a fridge, fills the agent's inventory with the fridge's ingredients; and *drop*, which, at the pot, empties the agent's inventory into the pot. Episodes consist of 20 timesteps and the agent receives $-0.1$ reward at each timestep until the goal, described below, is met (details in Appendix B.1; qualitative results in Appendix B.2).

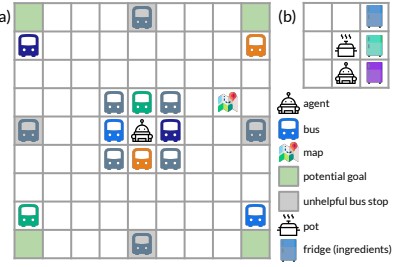

Figure 4: Didactic grid worlds to stress test exploration. (a) Navigation. (b) Cooking.

**Targeted exploration.** We first test if these methods can efficiently explore in the presence of distractions in two versions of the benchmark in Figure 4a: *distracting bus* and *map*. In both, there are 4 possible goals (the 4 green locations). During each episode, a goal is randomly sampled. Reaching the goal yields +1 reward and ends the episode. The 4 colored buses each lead to near a different potential green goal location when ridden and in different problems $\mu$, their destinations are set to be 1 of the 4! different permutations. The *distracting bus* version tests if the agent can ignore distractions by including unhelpful gray buses, which are never needed to optimally reach any goal. In different

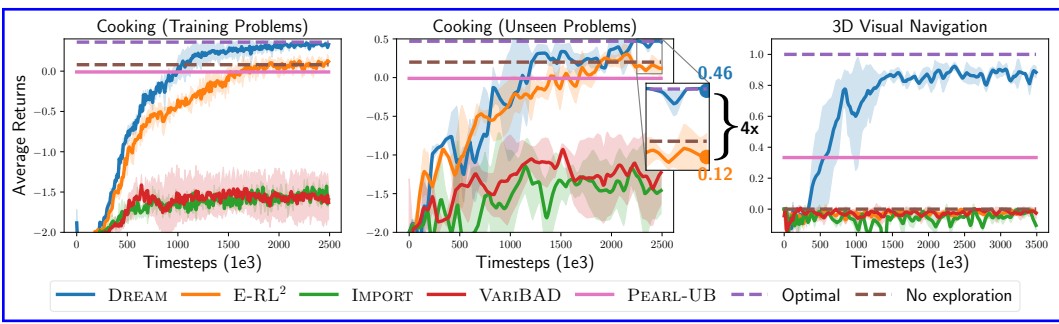

Figure 6: Cooking results: only DREAM achieves optimal reward on training problems (left) logged every 10 meta-training trials, and on generalizing to unseen problems (middle). 3D visual navigation results: only DREAM reads the sign and solves the task (right).

problems, the gray buses lead to different permutations of the gray locations. The *map* version tests if the agent can leverage objects for exploration by including a map that reveals the destinations of the colored buses when touched.

Figure 5 shows the results after 1M steps. DREAM learns to optimally explore and thus receives optimal reward in both versions: In *distracting bus*, it ignores the unhelpful gray buses and learns the destinations of all helpful buses by riding them. In *map*, it learns to leverage informative objects, by visiting the map and ending the exploration episode. During exploitation, DREAM immediately reaches the goal by riding the correct colored bus. In contrast, IMPORT and E-RL$^2$ get stuck in a local optimum, indicative of the coupling problem (Section 4.1), which achieves the same returns as no exploration at all. They do not explore the helpful buses or map and consequently sub-optimally exploit by just walking to the goal. VARIBAD learns slower, likely because it learns a dynamics model, but eventually matches the sub-optimal returns of IMPORT and RL$^2$ in ~3M steps (not shown).

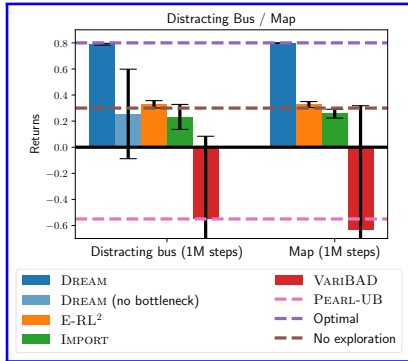

Figure 5: Navigation results. Only DREAM optimally explores all buses and the map.

PEARL achieves sub-optimal returns, even with infinite meta-training (see line for PEARL-UB), as follows. TS explores by sampling a problem ID from its posterior and executing its policy conditioned on this ID. Since for any given problem (bus configuration) and goal, the optimal problem-specific policy rides the one bus leading to the goal, TS does not explore optimally (i.e., explore all the buses or read the map), even with the optimal problem-specific policy and true posterior problem distribution.

Recall that DREAM tries to remove extraneous information from the problem ID with an information bottleneck that minimizes the mutual information $I(z; \mu)$ between problem IDs and the encoder $F_\psi(z \mid \mu)$. In *distracting bus*, we test the importance of the information bottleneck by ablating it from DREAM. As seen in Figure 5 (left), this ablation (DREAM (no bottleneck)) wastes its exploration on the gray unhelpful buses, since they are part of the problem, and consequently achieves low returns.

**Generalization to new problems.** We test generalization to unseen problems in a cooking benchmark (Figure 4b). The fridges on the right each contain 1 of 4 different (color-coded) ingredients, determined by the problem ID. The fridges' contents are unobserved until the agent uses the "pickup" action at the fridge. Goals (recipes) specify placing 2 correct ingredients in the pot in the right order. The agent receives positive reward for picking up and placing the correct ingredients, and negative reward for using the wrong ingredients. We hold out 1 of the $4^3 = 64$ problems from meta-training.

Figure 6 shows the results on training (left) and held-out (middle) problems. Only DREAM achieves near-optimal returns on both. During exploration, it investigates each fridge with the "pick up" action, and then directly retrieves the correct ingredients during exploitation. E-RL$^2$ gets stuck in a local optimum, only sometimes exploring the fridges. This achieves 3.8x lower returns, only slightly higher than no exploration at all. Here, leveraging the problem ID actually hurts IMPORT compared to E-RL$^2$. IMPORT successfully solves the task, given access to the problem ID, but fails without it. As before, VARIBAD learns slowly and TS (PEARL-UB) cannot learn optimal exploration.

## 6.2 SPARSE-REWARD 3D VISUAL NAVIGATION

We conclude with a challenging benchmark testing both sophisticated exploration and scalability to pixel inputs. We modify a benchmark from Kamienny et al. (2020) to increase both the exploration and scalability challenge by including more objects and a visual sign, illustrated in Figure 7. In the 3 different problems, the sign on the right says "blue", "red" or "green." The goals specify whether the agent should collect the key or block. The agent receives +1 reward for collecting the correct object (color specified by the sign, shape specified by the goal), -1 reward for the wrong object, and 0 reward otherwise. The agent begins the episode on the far side of the barrier and must walk around the barrier to visually "read" the sign. The agent's observations are

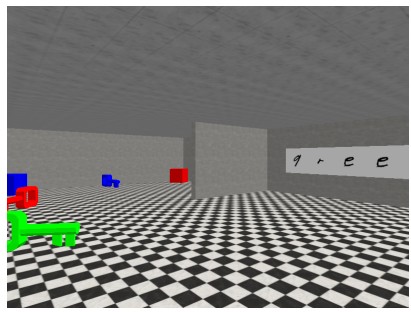

Figure 7: 3D Visual Navigation. The agent must read the sign to determine what colored object to go to.

$80 \times 60$ RGB images and its actions are to rotate left or right, move forward, or end the episode.

DREAM is the only method that learns to read the sign and achieve reward (Figure 6 right). All end-to-end approaches get stuck in local optima due to the chicken-and-egg coupling problem, where they do not learn to read the sign and hence stay away from all the objects, in fear of receiving negative reward. This achieves close to 0 returns, consistent with the results in Kamienny et al. (2020). As before, PEARL-UB cannot learn optimal exploration.

## 7 CONCLUSION

In summary, this work identifies a chicken-and-egg problem that end-to-end meta-RL approaches suffer from, where learning good exploitation requires already having learned good exploration and vice-versa. This creates challenging local optima, since typically neither exploration nor exploitation is good at the beginning of meta-training. We show that appropriately leveraging simple one-hot problem IDs allows us to break this cyclic dependency with DREAM. Consequently, DREAM has strong empirical performance on meta-RL problems requiring complex exploration, as well as substantial theoretical sample complexity improvements in the tabular setting. Though prior works also leverage the problem ID and use decoupled objectives that avoid the chicken-and-egg problem, no other existing approaches can recover optimal exploration empirically and theoretically like DREAM.

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

## A   DREAM TRAINING DETAILS

Algorithm 1 summarizes a practical algorithm for training DREAM. We parametrize both the exploration and exploitation policies as recurrent deep dueling double-Q networks (Wang et al., 2016b; van Hasselt et al., 2016), with exploration Q-values $\hat{Q}^{\text{exp}}(s, \tau^{\text{exp}}, a; \phi)$ parametrized by $\phi$ (and target network parameters $\phi'$) and exploitation Q-values $\hat{Q}^{\text{task}}(s, z, a; \theta)$ parametrized by $\theta$ (and target network parameters $\theta'$). We train on trials with one exploration and one exploitation episode, but can test on arbitrarily many exploitation episodes, as the exploitation policy acts on each episode independently (i.e. it does not maintain a hidden state across episodes). Using the choices for $F_\psi$ and $q_\omega$ in Section 4.3, training proceeds as follows.

We first sample a new problem for the trial and roll-out the exploration policy, adding the roll-out to a replay buffer (lines 7-9). Then, we roll-out the exploitation policy, adding the roll-out to a separate replay buffer (lines 10-12). We train the exploitation policy on both stochastic encodings of the problem ID $\mathcal{N}(f_\psi(\mu), \rho^2 I)$ and on encodings of the exploration trajectory $g_\omega(\tau^{\text{exp}})$.

Next, we sample from the replay buffers and update the parameters. First, we sample $(s_t, a_t, s_{t+1}, \mu, \tau^{\text{exp}})$-tuples from the exploration replay buffer and perform a normal DDQN update on the exploration Q-value parameters $\phi$ using rewards computed from the decoder (lines 13-15). Concretely, we minimize the following standard DDQN loss function w.r.t., the parameters $\phi$, where the rewards are computed according to Equation 4:

$$\mathcal{L}_{\text{exp}}(\phi) = \mathbb{E}\left[\left\|\hat{Q}^{\text{exp}}(s_t, \tau^{\text{exp}}_{:t-1}, a_t; \phi) - (r^{\text{exp}}_t + \gamma \hat{Q}^{\text{exp}}(s_{t+1}, [\tau^{\text{exp}}_{:t-1}; a_t; s_t], a_{\text{DDQN}}; \phi'))\right\|_2^2\right],$$

$$\text{where } r^{\text{exp}}_t = \left\|f_\psi(\mu) - g_\omega(\tau^{\text{exp}}_{:t})\right\|_2^2 - \left\|f_\psi(\mu) - g_\omega(\tau^{\text{exp}}_{:t-1})\right\|_2^2 - c$$

$$\text{and } a_{\text{DDQN}} = \arg\max_a \hat{Q}^{\text{exp}}(s_{t+1}, [\tau^{\text{exp}}_{:t-1}; a_t; s_t]; \phi).$$

We perform a similar update with the exploitation Q-value parameters (lines 16-17). We sample $(s, a, r, s', \mu, \tau^{\text{exp}})$-tuples from the exploitation replay buffer and perform a DDQN update from the encoding of the problem ID by minimizing the following loss:

$$\mathcal{L}_{\text{task-id}}(\theta, \psi) = \mathbb{E}\left[\left\|\hat{Q}^{\text{task}}(s, f_\psi(\mu), a; \theta) - (r + \hat{Q}^{\text{task}}(s', f_{\psi'}(\mu), a_{\text{prob}}; \theta'))\right\|_2^2\right],$$

$$\text{where } a_{\text{traj}} = \arg\max_a \hat{Q}^{\text{task}}(s', g_\omega(\tau^{\text{exp}}), a; \theta) \text{ and } a_{\text{prob}} = \arg\max_a \hat{Q}^{\text{task}}(s', f_\psi(\mu), a; \theta).$$

Finally, from the same exploitation replay buffer samples, we also update the problem ID embedder to enforce the information bottleneck (line 18) and the decoder to approximate the true conditional distribution (line 19) by minimizing the following losses respectively:

$$\mathcal{L}_{\text{bottleneck}}(\psi) = \mathbb{E}_\mu\left[\min\left(\|f_\psi(\mu)\|_2^2, K\right)\right]$$

$$\text{and } \mathcal{L}_{\text{decoder}}(\omega) = \mathbb{E}_{\tau^{\text{exp}}}\left[\sum_t \left\|f_\psi(\mu) - g_\omega(\tau^{\text{exp}}_{:t})\right\|_2^2\right].$$

Since the magnitude $\|f_\psi(\mu)\|_2^2$ partially determines the scale of the reward, we add a hyperparameter $K$ and only minimize the magnitude when it is larger than $K$. Altogether, we minimize the following loss:

$$\mathcal{L}(\phi, \theta, \omega, \psi) = \mathcal{L}_{\text{exp}}(\phi) + \mathcal{L}_{\text{task-id}}(\theta, \psi) + \mathcal{L}_{\text{bottleneck}}(\psi) + \mathcal{L}_{\text{decoder}}(\omega).$$

As is standard with deep Q-learning (Mnih et al., 2015), instead of sampling from the replay buffers and updating after each episode, we sample and perform all of these updates every 4 timesteps. We periodically update the target networks (lines 20-22).

## B   EXPERIMENT DETAILS

### B.1   PROBLEM DETAILS

**Distracting bus / map.** Riding each of the four colored buses teleports the agent to near one of the green goal locations in the corners. In different problems, the destinations of the colored buses

---

**Algorithm 1** DREAM DDQN

---

1: **Initialize** exploitation replay buffer $\mathcal{B}_{\text{task}} = \{\}$ and exploration replay buffer $\mathcal{B}_{\text{exp}} = \{\}$
2: **Initialize** exploitation Q-value $\hat{Q}^{\text{task}}$ parameters $\theta$ and target network parameters $\theta'$
3: **Initialize** exploration Q-value $\hat{Q}^{\text{exp}}$ parameters $\phi$ and target network parameters $\phi'$
4: **Initialize** problem ID embedder $f_\psi$ parameters $\psi$ and target parameters $\psi'$
5: **Initialize** trajectory embedder $g_\omega$ parameters $\omega$ and target parameters $\omega'$
6: **for** trial $= 1$ **to** max trials **do**
7:     Sample problem $\mu \sim p(\mu)$, defining MDP $\langle \mathcal{S}, \mathcal{A}, \mathcal{R}_\mu, T_\mu \rangle$
8:     Roll-out $\epsilon$-greedy exploration policy $\hat{Q}^{\text{exp}}(s_t, \tau_{:t}^{\text{exp}}, a_t; \phi)$, producing trajectory $\tau^{\text{exp}} = (s_0, a_0, \ldots, s_T)$.
9:     Add tuples to the exploration replay buffer $\mathcal{B}_{\text{exp}} = \mathcal{B}_{\text{exp}} \cup \{(s_t, a_t, s_{t+1}, \mu, \tau^{\text{exp}})\}_t$.

10:     Compute embedding $z \sim \mathcal{N}(f_\psi(\mu), \rho^2 I)$.
11:     Roll-out $\epsilon$-greedy exploitation policy $\hat{Q}^{\text{task}}(s_t, z, a_t; \theta)$, producing trajectory $(s_0, a_0, r_0, \ldots)$ with $r_t = \mathcal{R}_\mu(s_{t+1})$.
12:     Add tuples to the exploitation replay buffer $\mathcal{B}_{\text{task}} = \mathcal{B}_{\text{task}} \cup \{(s_t, a_t, r_t, s_{t+1}, \mu, \tau^{\text{exp}})\}_t$.

13:     Sample batches of $(s_t, a_t, s_{t+1}, \mu, \tau^{\text{exp}}) \sim \mathcal{B}_{\text{exp}}$ from exploration replay buffer.
14:     Compute reward $r_t^{\text{exp}} = \left\| f_\psi(\mu) - g_\omega(\tau_{:t}^{\text{exp}}) \right\|_2^2 - \left\| f_\psi(\mu) - g_\omega(\tau_{:t-1}^{\text{exp}}) \right\|_2^2 - c$ (Equation 4).
15:     Optimize $\phi$ with DDQN update with tuple $(s_t, a_t, r_t^{\text{exp}}, s_{t+1})$ with $\mathcal{L}_{\text{exp}}(\phi)$

16:     Sample batches of $(s, a, r, s', \mu, \tau^{\text{exp}}) \sim \mathcal{B}_{\text{task-id}}$ from exploitation replay buffer.
17:     Optimize $\theta$ and $\psi$ with DDQN update with tuple $((s, \mu), a, r, (s', \mu))$ with $\mathcal{L}_{\text{task-id}}(\theta, \psi)$
18:     Optimize $\psi$ on $\mathcal{L}_{\text{bottleneck}}(\psi) = \nabla_\psi \min(\left\| f_\psi(\mu) \right\|_2^2, K)$
19:     Optimize $\omega$ on $\mathcal{L}_{\text{decoder}}(\omega) = \nabla_\omega \sum_t \left\| f_\psi(\mu) - g_\omega(\tau_{:t}^{\text{exp}}) \right\|_2^2$ (Equation 3)

20:     **if** trial $\equiv 0 \pmod{\text{target freq}}$ **then**
21:         Update target parameters $\phi' = \phi$, $\theta' = \theta$, $\psi' = \psi$, $\omega' = \omega$
22:     **end if**
23: **end for**

---

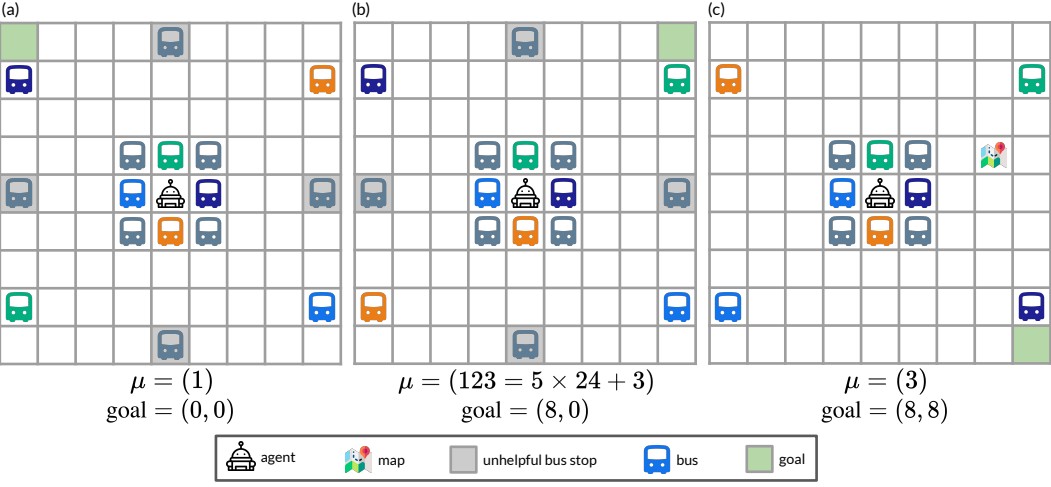

Figure 8: Examples of different *distracting bus* and *map* problems. (a) An example distracting bus problem. Though all unhelpful distracting buses are drawn in the same color (gray), the destinations of the gray buses are fixed within a problem. (b) Another example distracting bus problem. The destinations of the helpful colored buses are a different permutation (the orange and green buses have swapped locations). This takes on permutation $3 \equiv \mu \pmod{4!}$, instead of 1. The unhelpful gray buses are also a different permutation (not shown), taking on permutation $5 = \left\lfloor \frac{\mu}{4!} \right\rfloor$. (c) An example map problem. Touching the map reveals the destinations of the colored buses, by adding $\mu$ to the state observation.

change, but the bus positions and their destinations are fixed within each problem. Additionally, in the distracting bus domain, the problem ID also encodes the destinations of the gray buses, which are permutations of the four gray locations on the midpoints of the sides. More precisely, the problem ID $\mu \in \{0, 1, \ldots, 4! \times 4! = 576\}$ encodes both the permutation of the colored helpful bus destinations, indexed as $\mu \pmod{4!}$ and the permutation of the gray unhelpful bus destinations as $\lfloor \frac{\mu}{4!} \rfloor$. We hold out most of the problem IDs during meta-training ($\frac{23}{24} \times 576 = 552$ are held-out for meta-training).

In the map domain, the problem $\mu$ is an integer representing which of the 4! permutations of the four green goal locations the colored buses map to. The states include an extra dimension, which is set to 0 when the agent is not at the map, and is set to this integer value $\mu$ when the agent is at the map. Figure 8 displays three such examples.

**Cooking.** In different problems, the (color-coded) fridges contain 1 of 4 different ingredients. The ingredients in each fridge are unknown until the goes to the fridge and uses the pickup action. Figure 9 displays three example problems. The problem ID $\mu$ is an integer between 0 and $4^3$, where $\mu = 4^2 a + 4b + c$ indicates that the top right fridge has ingredient $a$, the middle fridge has ingredient $b$ and the bottom right fridge has ingredient $c$.

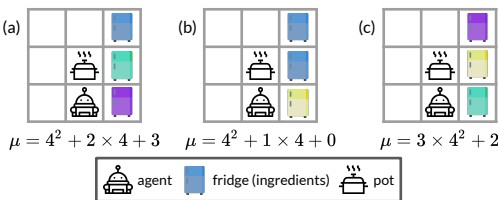

Figure 9: Three example cooking problems. The contents of the fridges (color-coded) are different in different problems.

The goals correspond to a recipe of placing the two correct ingredients in the pot in the right order. Goals are tuples $(a, b)$, which indicate placing ingredient $a$ in the pot first, followed by ingredient $b$. In a given problem, we only sample goals involving the recipes actually present in that problem. During meta-training, we hold out a randomly selected problem $\mu = 11$.

We use the following reward function $\mathcal{R}_\mu$. The agent receives a per timestep penalty of $-0.1$ reward and receives $+0.25$ reward for completing each of the four steps: (i) picking up the first ingredient specified by the goal; (ii) placing the first ingredient in the pot; (iii) picking up the second ingredient specified by the goal; and (iv) placing the second ingredient in the pot. To prevent the agent from gaming the reward function, e.g., by repeatedly picking up the first ingredient, dropping the first ingredient anywhere but in the pot yields a penalty of $-0.25$ reward, and similarly for all steps. To encourage efficient exploration, the agent also receives a penalty of $-0.25$ reward for picking up the wrong ingredient.

**Cooking without goals.** While we evaluate on goal-conditioned benchmarks to deepen the exploration challenge, forcing the agent to discover all the relevant information for *any* potential goal, many standard benchmarks (Finn et al., 2017; Yu et al., 2019) don't involve goals. We therefore include a variant of the cooking task, where there are no goals. We simply concatenate the goal (recipe) to the problem ID $\mu$. Additionally, we modify the rewards so that picking up the second ingredient yields $+0.25$ and dropping it yields $-0.25$ reward, so that it is possible to infer the recipe from the rewards. Finally, to make the problem harder, the agent cannot pick up new ingredients unless its inventory is empty (by using the drop action), and we also increase the number of ingredients to 7. The results are in Section B.2.

**Sparse-reward 3D visual navigation.** We implement this domain in Gym MiniWorld (Chevalier-Boisvert, 2018), where the agent's observations are $80 \times 60 \times 3$ RGB arrays. There are three problems $\mu = 0$ (the sign says "blue"), $\mu = 1$ (the sign says "red"), and $\mu = 2$ (the sign says "green"). There are two goals, represented as 0 and 1, corresponding to picking up the key and the box, respectively. The reward function $\mathcal{A}_\mu(s, i)$ is +1 for picking up the correct colored object (according to $\mu$) and the correct type of object (according to the goal) and $-1$ for picking up an object of the incorrect color or type. Otherwise, the reward is 0. On each episode, the agent begins at a random location on the other side of the barrier from the sign.

## B.2 ADDITIONAL RESULTS

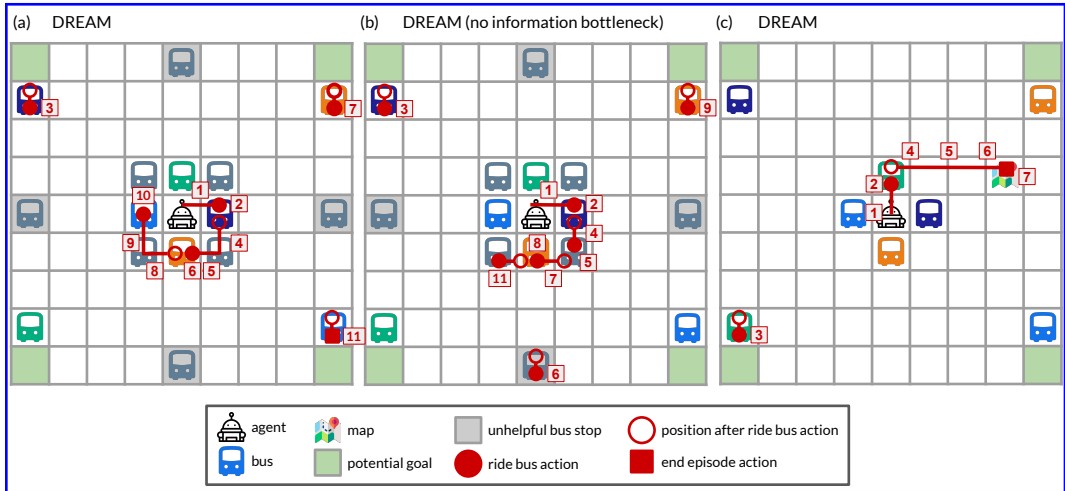

Figure 10: Examples of DREAM's learned exploration behavior. (a) DREAM learns the optimal exploration behavior on the *distraction* variant: riding 3 of the 4 helpful colored buses, which allows it to infer the destinations of all colored buses and efficiently reach any goal during exploitation episodes. (a) Without the information bottleneck, DREAM also explores the unhelpful gray buses, since they are part of the problem. This wastes exploration steps, and leads to lower returns during exploitation episodes. (c) DREAM learns the optimal exploration on the *map* variant: it goes to read the map revealing all the buses' destinations, and then ends the episode, though it unnecessarily rides one of the buses.

**Analysis of the learned policies.** Please see https://anonymouspapersubmission.github.io/dream/ for videos and analysis of the exploration and exploitation behavior learned by DREAM and other approaches, which is described in text below.

**Distracting bus / map.** Figure 10 shows the exploration policy DREAM learns on the distracting bus and map domains. With the information bottleneck, DREAM optimally explores by riding 3 of the 4 colored buses and inferring the destination of the last colored bus (Figure 8a). Without the information bottleneck, DREAM explores the unhelpful gray buses and runs out of time to explore all of the colored buses, leading to lower reward (Figure 8b). In the map domain, DREAM optimally explores by visiting the map and terminating the exploration episode. In contrast, the other methods (RL$^2$, IMPORT, VARIBAD) rarely visit the colored buses or map during exploration and consequently walk to their destination during exploitation, which requires more timesteps and therefore receives lower returns.

In Figure 11, we additionally visualize the exploration trajectory encodings $g_\omega(\tau^{\text{exp}})$ and problem ID encodings $f_\psi(\mu)$ that DREAM learns in the distracting bus domain by applying t-SNE (van der Maaten & Hinton, 2008). We visualize the encodings of all possible problem IDs as dots. They naturally cluster into 4! = 24 clusters, where the problems within each cluster differ only in the destinations of the gray distracting buses, and not the colored buses. Problems in the support of the true posterior $p(\mu \mid \tau^{\text{exp}})$ are drawn in green, while problems outside the support (e.g., a problem that specifies that riding the green bus goes to location $(0, 1)$ when it has already been observed in $\tau^{\text{exp}}$ that riding the orange bus goes to location $(0, 1)$) are drawn in red. We also plot the encoding of the exploration trajectory $\tau^{\text{exp}}$ so far as a blue cross and the mean of the green clusters as a black square. We find that the encoding of the exploration trajectory $g_\omega(\tau^{\text{exp}})$ tracks the mean of the green clusters until the end of the exploration episode, when only one cluster remains, and the destinations of all the colored buses has been discovered. Intuitively, this captures uncertainty in what the potential problem ID may be. More precisely, when the decoder is a Gaussian, placing $g_\omega(\tau^{\text{exp}})$ at the center of the encodings of problems in the support exactly minimizes Equation 3.

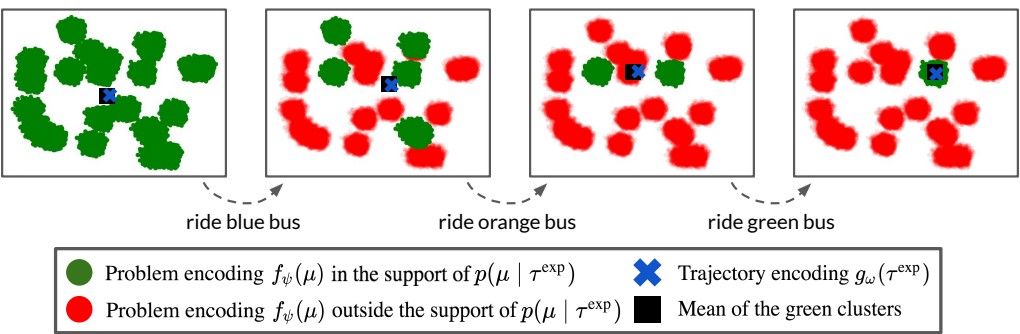

Figure 11: DREAM's learned encodings of the exploration trajectory and problems visualized with t-SNE (van der Maaten & Hinton, 2008).

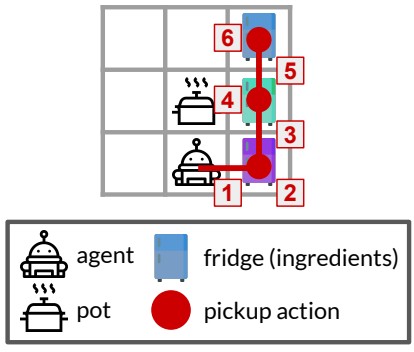

Figure 12: DREAM learns the optimal exploration policy, which learns the fridges' contents by going to each fridge and using the pickup action.

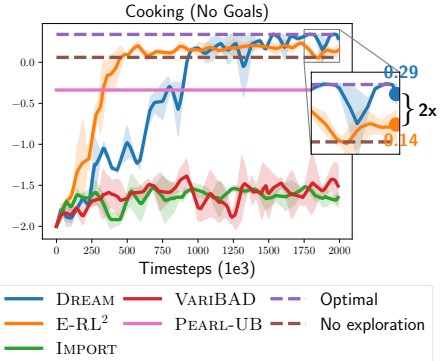

Figure 13: Cooking without goals results. Only DREAM learns the optimal policy, achieving ~2x more reward than the next best approach.

**Robustness to imperfections in the problem ID.** Recall that the problem ID is a simple and easy-to-provide unique one-hot for each problem. We test of DREAM is robust to imperfections in the problem ID by assigning each problem in the *map* benchmark 3 different problem IDs. When a problem is sampled during meta-training, it is randomly labeled with 1 of these 3 different problem IDs. We find that this imperfection in the problem ID does not impact DREAM's final performance at all: it still achieves optimal exploitation returns of 0.8 after 1M time steps of training.

**Cooking.** Figure 12 shows the exploration policy DREAM learns on the cooking domain, which visits each of the fridges and investigates the contents with the "pickup" action. In contrast, the other methods rarely visit the fridges during exploration, and instead determine the locations of the ingredients during exploitation, which requires more timesteps and therefore receives lower returns.

**Cooking without goals.** We provide additional results in the case where the cooking domain is modified to not include goals (see Section B.1). The results are summarized in Figure 13 and show the same trends as the results in original cooking benchmark. DREAM learns to optimally explore by investigating the fridges, and then also optimally exploits, by directly collecting the relevant ingredients. The next best approach E-RL$^2$, only sometimes explores the fridges, again getting stuck in a local optimum, yielding only slightly higher reward than no exploration at all.

**Sparse-reward 3D visual navigation.** DREAM optimally explores by walking around the barrier and reading the sign. See https://anonymouspapersubmission.github.io/dream/ for videos. The other methods do not read the sign at all and therefore cannot solve the problem.

### B.3    OTHER APPROACHES AND ARCHITECTURE DETAILS

In this section, we detail the loss functions that E-RL$^2$, IMPORT, and VARIBAD optimize, as well as the model architectures used in our experiments. Where possible, we use the same model architecture for all methods: DREAM, E-RL$^2$, IMPORT, and VARIBAD. All approaches are implemented in PyTorch (Paszke et al., 2017), using a DQN implementation adapted from Liu et al. (2020b) and code adapted from Liu et al. (2020a).

**State and problem ID embeddings.**    All approaches use the same method to embed the state and problem ID. For these embeddings, we embed each dimension independently with an embedding matrix of output dimension 32. Then, we concatenate the per-dimension embeddings and apply two linear layers with output dimensions 256 and 64 respectively, with ReLU activations.

In the 3D visual navigation task, we use a different embedding scheme for the states, as they are images. We apply 3 CNN layers, each with 32 output layers and stride length 2, and with kernel sizes of 5, 5, and 4 respectively. We apply ReLU activations between the CNN layers and apply a final linear layer to the flattened output of the CNN layers, with an output dimension of 128.

All state and problem ID embeddings below use this scheme.

**Experience embeddings.**    E-RL$^2$, IMPORT, VARIBAD and the exploration policy in DREAM also learn an embedding of the history of prior experiences $\tau^{\text{exp}} = (s_0, a_0, r_0, s_1, \ldots)$ and current state $s_T$. To do this, we first separately embed each $(s_{t+1}, a_t, r_t, d_t)$-tuple, where $d_t$ is an episode termination flag (true if the episode ends on this experience, and false otherwise), as follows:

- Embed the state $s_t$ as $e(s_t)$, using the state embedding scheme described above.

- Embed the action $a_t$ as $e(a_t)$ with an embedding matrix of output dimension 16. We set $a_{-1}$ to be 0.

- For the standard meta-RL setting and during exploitation episodes, embed the rewards with a linear layer of output dimension 16. With reward-free exploration in IMRL, the rewards are not embedded in the exploration policy of DREAM, and are embedded as 0 reward for the other approaches, since the same policy is used during both exploration and exploitation. We set $r_{-1}$ to be 0.

- Embed the episode termination $d_t$ as $e(d_t)$ with an embedding matrix of output dimension 16. Note that $d$ is true during all episode terminations within a trial for RL$^2$, IMPORT, and VARIBAD.

Then, we apply a final linear layer with output dimension 64 to the concatenation of the above $[e(s_t); e(a_t); e(r_t); d_t]$. Finally, to obtain an embedding of the entire history $\tau^{\text{exp}}$, we embed each experience separately as above, and then pass an LSTM with hidden dimension 64 over the experience embeddings, where the initial hidden and cell states are set to be 0-vectors.

**DREAM.**    For the decoder $g_\omega(\tau^{\text{exp}} = (s_0, a_0, r_0, s_1, \ldots, s_T))$, we embed each transition $(s_t, a_t, r_t, s_{t+1})$ of the exploration trajectory $\tau^{\text{exp}}$ using the same embedding scheme as above, except we also embed the next state $s_{t+1}$. We do not embed the rewards in the reward-free adaptation version of IMRL. Then, given embeddings for each transition, we embed the entire trajectory by passing an LSTM with output dimension 128 on top of the transition embeddings, followed by two linear layers of output dimension 128 and 64 with ReLU activations.

For the exploitation policy Q-values $\hat{Q}_\theta^{\text{task}}(a \mid s, z)$, we either choose $z$ to be the decoder embedding of the exploration trajectory $g_\omega(\tau^{\text{exp}})$ or to be an embedding of the problem ID $e_\theta(\mu)$, where we always use the exploration trajectory embedding $g_\omega(\tau^{\text{exp}})$ at meta-test time. We embed the state with a learned embedding functions $e(s)$. Then we apply a linear layer of output dimension 64 to the concatenation of $[e(s); z]$ with a ReLU activation. Finally, we apply two linear layer heads of output dimension 1 and $|\mathcal{A}|$ respectively to form estimates of the value and advantage functions, using the dueling Q-network parametrization. To obtain Q-values, we add the value function to the advantage function, subtracting the mean of the advantages.

For the exploration policy Q-values $\hat{Q}_\phi^{\text{exp}}(a_t \mid s_t, \tau_{:t}^{\text{exp}})$, we embed the $s_t$ and $\tau_{:t}^{\text{exp}}$ according to the embedding scheme above. Then, we apply two linear layer heads to obtain value and advantage estimates as above.

**E-RL$^2$.** E-RL$^2$ learns a policy $\pi(a_t \mid s_t, \tau_{:t})$ producing actions $a_t$ given the state $s_t$ and history $\tau_{:t}$. Like with all approaches, we parametrize this with dueling double Q-networks, learning Q-values $\hat{Q}(s_t, \tau_{:t}, a_t)$. We embed the current state $s_t$ and history $\tau_{:t}$ using the embedding scheme described above (with episode termination embeddings). Then, we apply two final linear layer heads to obtain value and advantage estimates.

**IMPORT**  IMPORT also learns a recurrent policy $\pi(a_t \mid s_t, z)$, but conditions on the embedding $z$, which is either an embedding of the problem $\mu$ or the history $\tau_{:t}$. We also parametrize this policy with dueling double Q-networks, learning Q-values $\hat{Q}(s_t, z, a_t)$. We embed the state $s_t$ as $e(s_t)$, the problem $\mu$ as $e_\phi(\mu)$ and the history $\tau_{:t}$ as $e_\theta(\tau_{:t})$ using the previously described embedding schemes. Then we alternate meta-training trials between choosing $z = e_\phi(\mu)$ and $z = e_\theta(\tau_{:t})$. We apply a linear layer of output dimension 64 to the concatenation $[e(s_t); z]$ with ReLU activations and then apply two linear layer heads to obtain value and advantage estimates.

Additionally, IMPORT uses the following auxiliary loss function to encourage the history embedding $e_\theta(\tau_{:t})$ to be close to the problem embedding $e_\phi(\mu)$ (optimized only w.r.t., $\theta$):

$$\mathcal{L}_{\text{IMPORT}}(\theta) = \beta\mathbb{E}_{(\tau,\mu)}\left[\sum_t \|e_\theta(\tau_{:t}) - e_\phi(\mu)\|_2^2\right],$$

where $\tau$ is a trajectory from rolling out the policy on problem $\mu$. Following Kamienny et al. (2020), we use $\beta = 1$ in our final experiments, and found that performance changed very little when we experimented with other values of $\beta$.

**VARIBAD.**  VARIBAD also learns a recurrent policy $\pi(a_t \mid z)$, but over a *belief state* z capturing the history $\tau_{:t}$ and current state $s_t$. We also parametrize this dueling double Q-networks, learning Q-values $\hat{Q}(s_t, z, a_t)$.

VARIBAD encodes the belief state with an encoder $\text{enc}(z \mid s_t, \tau : t)$. Our implementation of this encoder embeds $s_t$ and $\tau_{:t}$ using the same experience embedding approach as above, and use the output as the mean $m$ for a distribution. Then, we set $\text{enc}(z \mid s_t, \tau : t) = \mathcal{N}(m, \nu^2 I)$, where $\nu^2 = 0.00001$. We also tried learning the variance instead of fixing it to $\nu^2 I$ by applying a linear head to the output of the experience embeddings, but found no change in performance, so stuck with the simpler fixed variance approach. Finally, after sampling $z$ from the encoder, we also embed the current state $s_t$ as $e(s_t)$ and apply a linear layer of output dimension 64 to the concatenation $[e(s_t); z]$. Then, we apply two linear layer heads to obtain value and advantage estimates.

VARIBAD does not update the encoder via gradients through the policy. Instead, VARIBAD jointly trains the encoder with state decoder $\hat{T}(s' \mid a, s, z)$ and reward decoder $\hat{\mathcal{R}}(s' \mid a, s, z)$, where $z$ is sampled from the encoder, as follows. Both decoders embed the action $a$ as $e(a)$ with an embedding matrix of output dimension 32 and embed the state $s$ as $e(s)$. Then we apply two linear layers with output dimension 128 to the concatenation $[e(s); e(a); z]$. Finally, we apply two linear heads, one for the state decoder and one for the reward decoder and take the mean-squared error with the true next state $s'$ and the true rewards $r$ respectively. In the 3D visual navigation domain, we remove the state decoder, because the state is too high-dimensional to predict. Note that Zintgraf et al. (2019) found better results when removing the state decoder in all experiments. We also tried to remove the state decoder in the grid world experiments, but found better performance when keeping the state decoder. We also found that VARIBAD performed better without the KL loss term, so we excluded that for our final experiments.

### B.4  HYPERPARAMETERS

In this section, we detail the hyperparameters used in our experiments. Where possible, we used the default DQN hyperparameter values from Mnih et al. (2015). and shared the same hyperparameter values across all methods for fairness. We optimize all methods with the Adam optimizer (**?**). Table 1 summarizes the shared hyperparameters used by all methods and we detail any differences in hyperparameters between the methods below.

| Hyperparameter | Value |
|---|---|
| Discount Factor $\gamma$ | 0.99 |
| Test-time $\epsilon$ | 0 |
| Learning Rate | 0.0001 |
| Replay buffer batch size | 32 |
| Target parameters syncing frequency | 5000 updates |
| Update frequency | 4 steps |
| Grad norm clipping | 10 |

Table 1: Hyperparameters shared across all methods: DREAM, RL$^2$, IMPORT, and VARIBAD.

All methods use a linear decaying $\epsilon$ schedule for $\epsilon$-greedy exploration. For RL$^2$, IMPORT, and VARIBAD, we decay $\epsilon$ from 1 to 0.01 over 500000 steps. For DREAM, we split the decaying between the exploration and exploitation policies. We decay each policy's $\epsilon$ from 1 to 0.01 over 250000 steps.

We train the recurrent policies (DREAM's exploration policy, RL$^2$, IMPORT, and VARIBAD) with a simplified version of the methods in Kapturowski et al. (2019) by storing a replay buffer with up to 16000 sequences of 50 consecutive timesteps. We decrease the maximum size from 16000 to 10000 for the 3D visual navigation experiments in order to fit inside a single NVIDIA GeForce RTX 2080 GPU. For DREAM's exploitation policy, the replay buffer stores up to 100K experiences (60K for 3D visual navigation).

For DREAM, we additionally use per timestep exploration reward penalty $c = 0.01$, decoder and stochastic encoder variance $\rho^2 = 0.1$, and information bottleneck weight $\lambda = 1$. Note that this information bottleneck weight $\lambda$ could be adapted via dual gradient descent to solve the constrained optimization problem in Appendix E, but we find that dynamically adjusting $\lambda$ is not necessary for good performance. For the MiniWorld experiments, we use $c = 0$.

## C ANALYSIS

### C.1 CONSISTENCY

We restate the consistency result of DREAM (Section 5.1) and prove it below.

**Proposition 1.** *Assume $\langle \mathcal{S}, \mathcal{A}, \mathcal{R}_\mu, \mathcal{T}_\mu \rangle$ is ergodic for all problems $\mu \in \mathcal{M}$. Let $V^*(\mu)$ be the maximum expected returns achievable by any exploitation policy with access to the problem ID $\mu$, i.e., with complete information. Let $\pi_\star^{task}, \pi_\star^{exp}, F_\star$ and $q_\star(z \mid \tau^{exp})$ be the optimizers of the DREAM objective. Then, if the function classes DREAM optimizes over are well-specified, there exists a finite $T$ such that if the length of the exploration episode is at least $T$,*

$$\mathbb{E}_{\mu \sim p(\mu), \tau^{exp} \sim \pi_\star^{exp}, z \sim q_\star(z|\tau^{exp})} \left[ V^{\pi_\star^{task}}(z; \mu) \right] = \mathbb{E}_{\mu \sim p(\mu)} \left[ V^*(\mu) \right].$$

*Proof.* Recall that $\pi_\star^{task}$ and $F_\star(z \mid \mu)$ are optimized with an information bottleneck according to Equation 2 in order to solve the constrained optimization problem in Appendix E. Note that if $\pi_\star^{task}$ is optimized over an expressive enough function class and $\lambda$ approaches 0, which is necessary to solve the constrained optimization problem associated with the information bottleneck (Appendix E), then $\pi_\star^{task}$ achieves the desired expected returns conditioned on the stochastic encoding of the problem $F_\star(z \mid \mu)$ (i.e., has complete information):

$$\mathbb{E}_{\mu \sim p(\mu), z \sim F_\star(z|\mu)} \left[ V^{\pi_\star^{task}}(z; \mu) \right] = \mathbb{E}_{\mu \sim p(\mu)} \left[ V^*(\mu) \right],$$

where $V^{\pi_\star^{task}}(z; \mu)$ is the expected returns of $\pi_\star^{task}$ on problem $\mu$ given embedding $z$. Therefore, it suffices to show that the distribution over $z$ from the decoder $q_\star(z \mid \tau^{exp})$ is equal to the distribution from the encoder $F_\star(z \mid \mu)$ for all exploration trajectories in the support of $\pi^{exp}(\tau^{exp} \mid \mu)^2$, for each problem $\mu$. Then,

$$\mathbb{E}_{\mu \sim p(\mu), \tau^{exp} \sim \pi_\star^{exp}, z \sim q_\star(z|\tau^{exp})} \left[ V^{\pi_\star^{task}}(z; \mu) \right] = \mathbb{E}_{\mu \sim p(\mu), z \sim F_\star(z|\mu)} \left[ V^{\pi_\star^{task}}(z; \mu) \right]$$

$$= \mathbb{E}_{\mu \sim p(\mu)} \left[ V^*(\mu) \right]$$

---

$^2$We slightly abuse notation to use $\pi^{exp}(\tau^{exp} \mid \mu)$ to denote the distribution of exploration trajectories $\tau^{exp}$ from rolling out $\pi^{exp}$ on problem $\mu$.

as desired. We show that this occurs as follows.

Given stochastic encoder $F_\star(z \mid \mu)$, exploration policy $\pi_\star^{\text{exp}}$ maximizes $I(\tau^{\text{exp}}; z) = H(z) - H(z \mid \tau^{\text{exp}})$ (Equation 3) by assumption. Since only $H(z \mid \tau^{\text{exp}})$ depends on $\pi_\star^{\text{exp}}$, the exploration policy outputs trajectories that minimize

$$H(z \mid \tau^{\text{exp}}) = \mathbb{E}_{\mu \sim p(\mu)} \left[ \mathbb{E}_{\tau^{\text{exp}} \sim \pi^{\text{exp}}(\tau^{\text{exp}} \sim \mu)} \left[ \mathbb{E}_{z \sim F_\star(z \mid \mu)} \left[ -\log p(z \mid \tau^{\text{exp}}) \right] \right] \right]$$

$$= \mathbb{E}_{\mu \sim p(\mu)} \left[ \mathbb{E}_{\tau^{\text{exp}} \sim \pi^{\text{exp}}(\tau^{\text{exp}} \sim \mu)} \left[ H(F_\star(z \mid \mu), p(z \mid \tau^{\text{exp}})) \right] \right],$$

where $p(z \mid \tau^{\text{exp}})$ is the true conditional distribution and $H(F_\star(z \mid \mu), p(z \mid \tau^{\text{exp}}))$ is the cross-entropy. The cross-entropy is minimized when $p(z \mid \tau^{\text{exp}}) = F_\star(z \mid \mu)$, which can be achieved with long enough exploration trajectories $T$ if $\langle \mathcal{S}, \mathcal{A}, \mathcal{R}_\mu, \mathcal{T}_\mu \rangle$ is ergodic (by visiting each transition sufficiently many times). Optimized over an expressive enough function class, $q_\star(z \mid \tau^{\text{exp}})$ equals the true conditional distribution $p(z \mid \tau^{\text{exp}})$ at the optimum of Equation 3, which equals $F_\star(z \mid \mu)$ as desired. $\qquad\square$

### C.2 Tabular Example

We first formally detail a more general form of the simple tabular example in Section 5.2, where episodes are horizon $H$ rather than 1-step bandit problems. Then we prove sample complexity bounds for RL$^2$ and DREAM, with $\epsilon$-greedy tabular Q-learning with $\epsilon = 1$, i.e., uniform random exploration.

**Setting.** We construct this horizon $H$ setting so that taking a *sequence* of $H$ actions $\mathbf{a}_\star$ (instead of a single action as before) in the exploration episode leads to a trajectory $\tau_\star^{\text{exp}}$ that reveals the problem $\mu$; all other action sequences $\mathbf{a}$ lead to a trajectory $\tau_{\mathbf{a}}^{\text{exp}}$ that reveals no information. Similarly, the problem $\mu$ identifies a unique sequence of $H$ actions $\mathbf{a}_\mu$ that receives reward 1 during exploitation, while all other action sequences receive reward 0. Again, taking the action sequence $\mathbf{a}_\star$ during exploration is therefore necessary and sufficient to obtain optimal reward 1 during exploitation.

We formally construct this setting by considering a family of episodic MDPs $\langle \mathcal{S}, \mathcal{A}, \mathcal{R}_\mu, T_\mu \rangle$ parametrized by the problem ID $\mu \in \mathcal{M}$, where:

- Each exploitation and exploration episode is horizon $H$.

- The action space $\mathcal{A}$ consists of $A$ discrete actions $\{1, 2, \ldots, A\}$.

- The space of problems $\mathcal{M} = \{1, 2, \ldots, |\mathcal{A}|^H\}$ and the distribution $p(\mu)$ is uniform.

To reveal the problem via the optimal action sequence $\mathbf{a}_\star$ and to allow $\mathbf{a}_\mu$ to uniquely receive reward, we construct the state space and deterministic dynamics as follows.

- States $s \in \mathcal{S}$ are $(H+2)$-dimensional and the deterministic dynamics are constructed so the first index represents the current timestep $t$, the middle $H$ indices represent the history of actions taken, and the last index reveals the problem ID if $\mathbf{a}_\star$ is taken. The initial state is the zero vector $s_0 = \mathbf{0}$ and we denote the state at the $t$-th timestep $s_t$ as $(t, a_0, a_1, \ldots, a_{t-1}, 0, \ldots, 0, 0)$.

- The optimal exploration action sequence $\mathbf{a}_\star$ is set to be taking action 1 for $H$ timesteps. In problem $\mu$ taking action $a_{H-1} = 1$ at state $s_{H-1} = (H-1, a_0 = 1, \ldots, a_{H-2} = 1, 0, 0)$ (i.e., taking the entire action sequence $\mathbf{a}_\star$) transitions to the state $s_H = (H, a_0 = 1, \ldots, a_{H-2} = 1, a_{H-1} = 1, \mu)$, revealing the problem $\mu$.

- The action sequence $\mathbf{a}_\mu$ identified by the problem $\mu$ is set as the problem $\mu$ in base $|\mathcal{A}|$: i.e., $\mathbf{a}_\mu$ is a sequence of $H$ actions $(a_0, a_1, \ldots, a_{H-1})$ with $\sum_{t=0}^{H-1} a_t |\mathcal{A}|^t = \mu$. In problem $\mu$ with $\mathbf{a}_\mu = (a_0, a_1, \ldots, a_{H-1})$, taking action $a_{H-1}$ at timestep $H-1$ at state $s_{H-1} = (H-1, a_0, a_1, \ldots, a_{H-2}, 0, 0)$ (i.e., taking the entire action sequence $\mathbf{a}_\mu$) yields $\mathcal{R}_\mu(s_{H-1}, a_{H-1}) = 1$. Reward is 0 everywhere else: i.e., $\mathcal{R}_\mu(s, a) = 0$ for all other states $s$ and actions $a$.

- With these dynamics, the exploration trajectory $\tau_{\mathbf{a}}^{\text{exp}} = (s_0, a_0, r_0, \ldots, s_H)$ is uniquely identified by the action sequence $\mathbf{a}$ and the problem $\mu$ if revealed in $s_H$. We therefore write $\tau_{\mathbf{a}}^{\text{exp}} = (\mathbf{a}, \mu)$ for when $\mathbf{a} = \mathbf{a}_\star$ reveals the problem $\mu$, and $\tau_{\mathbf{a}}^{\text{exp}} = (\mathbf{a}, 0)$, otherwise.

**Uniform random exploration.** In this general setting, we analyze the number of samples required to learn the optimal exploration policy with $RL^2$ and DREAM via $\epsilon$-greedy tabular Q-learning. We formally analyze the simpler case where $\epsilon = 1$, i.e., uniform random exploration, but empirically find that DREAM only learns faster with smaller $\epsilon$, and $RL^2$ only learns slower.

In this particular tabular example with deterministic dynamics that encode the entire action history and rewards, learning a per timestep Q-value is equivalent to learning a Q-value for the entire trajectory. Hence, we denote exploration Q-values $\hat{Q}^{\text{exp}}(\mathbf{a})$ estimating the returns from taking the entire sequence of $H$ actions $\mathbf{a}$ at the initial state $s_0$ and exeuction Q-values $\hat{Q}^{\text{task}}(\tau^{\text{exp}}, \mathbf{a})$ estimating the returns from taking the entire sequence of $H$ actions $\mathbf{a}$ at the initial state $s_0$ given the exploration trajectory $\tau^{\text{exp}}$. We drop $s_0$ from notation, since it is fixed.

Recall that $RL^2$ learns exploration Q-values $\hat{Q}^{\text{exp}}$ by regressing toward the exploitation Q-values $\hat{Q}^{\text{task}}$. We estimate the exploitation Q-values $\hat{Q}^{\text{task}}(\tau^{\text{exp}}, \mathbf{a})$ as the sample mean of returns from taking actions $\mathbf{a}$ given the exploration trajectory $\tau^{\text{exp}}$ and estimate the exploration Q-values $\hat{Q}^{\text{exp}}(\mathbf{a})$ as the sample mean of the targets. More precisely, for action sequences $\mathbf{a} \neq \mathbf{a}_\star$, the resulting exploration trajectory $\tau_{\mathbf{a}}^{\text{exp}}$ is deterministically $(\mathbf{a}, 0)$, so we set $\hat{Q}^{\text{exp}}(\mathbf{a}) = \hat{V}^{\text{task}}(\tau_{\mathbf{a}}^{\text{exp}}) = \max_{\mathbf{a}'} \hat{Q}^{\text{task}}(\tau_{\mathbf{a}}^{\text{exp}}, \mathbf{a}')$. For $\mathbf{a}_\star$, the resulting exploration trajectory $\tau_{\mathbf{a}_\star}^{\text{exp}}$ may be any of $(\mathbf{a}_\star, \mu)$ for $\mu \in \mathcal{M}$, so we set $\hat{Q}^{\text{exp}}(\mathbf{a}_\star)$ as the empirical mean of $\hat{V}^{\text{task}}(\tau_{\mathbf{a}_\star}^{\text{exp}})$ of observed $\tau_{\mathbf{a}_\star}^{\text{exp}}$.

Recall that DREAM learns exploration Q-values $\hat{Q}^{\text{exp}}$ by regressing toward the learned decoder $\log \hat{q}(\mu \mid \tau_{\mathbf{a}}^{\text{exp}})$. We estimate the decoder $\hat{q}(\mu \mid \tau_{\mathbf{a}}^{\text{exp}})$ as the empirical counts of $(\mu, \tau_{\mathbf{a}}^{\text{exp}})$ divided by the empirical counts of $\tau_{\mathbf{a}}^{\text{exp}}$ and similarly estimate the Q-values as the empirical mean of $\log \hat{q}(\mu \mid \tau_{\mathbf{a}}^{\text{exp}})$. We denote the exploration Q-values learned after $t$ timesteps as $\hat{Q}_t^{\text{exp}}$, and similarly denote the estimated decoder after $t$ timesteps as $\hat{q}_t$.

Given this setup, we are ready to state the formal sample complexity results. Intuitively, learning the exploitation Q-values for $RL^2$ is slow, because, in problem $\mu$, it involves observing the optimal exploration trajectory from taking actions $\mathbf{a}_\star$ and then observing the corresponding exploitation actions $\mathbf{a}_\mu$, which only jointly happens roughly once per $|\mathcal{A}|^{2H}$ samples. Since $RL^2$ regresses the exploration Q-values toward the exploitation Q-values, the exploration Q-values are also slow to learn. In contrast, learning the decoder $\hat{q}(\mu \mid \tau_{\mathbf{a}}^{\text{exp}})$ is much faster, as it is independent of the exploitation actions, and in particular, already learns the correct value from a single sample of $\mathbf{a}_\star$. We formalize this intuition in the following proposition, which shows that DREAM learns in a factor of at least $|\mathcal{A}|^H |\mathcal{M}|$ fewer samples than $RL^2$.

**Proposition 2.** *Let $T$ be the number of samples from uniform random exploration such that the greedy-exploration policy is guaranteed to be optimal (i.e., $\arg\max_{\mathbf{a}} \hat{Q}_t^{exp}(\mathbf{a}) = \mathbf{a}_\star$) for all $t \geq T$. If $\hat{Q}^{exp}$ is learned with DREAM, the expected value of $T$ is $\mathcal{O}(|\mathcal{A}|^H \log |\mathcal{A}|^H)$. If $\hat{Q}^{exp}$ is learned with $RL^2$, the expected value of $T$ is $\Omega(|\mathcal{A}|^{2H} |\mathcal{M}| \log |\mathcal{A}|^H)$.*

*Proof.* For DREAM, $\hat{Q}_T^{\text{exp}}(\mathbf{a}_\star) > \hat{Q}_T^{\text{exp}}(\mathbf{a})$ for all $\mathbf{a} \neq \mathbf{a}_\star$ if $\log \hat{q}_T(\mu \mid (\mathbf{a}_\star, \mu)) > \log \hat{q}_T(\mu \mid (\mathbf{a}, 0))$ for all $\mu$ and $\mathbf{a} \neq \mathbf{a}_\star$. For all $t \geq T$, $\hat{Q}_t^{\text{exp}}$ is guaranteed to be optimal, since no sequence of samples will cause $\log \hat{q}_t(\mu \mid (\mathbf{a}_\star, \mu)) = 0 \leq \log \hat{q}_t(\mu \mid (\mathbf{a}, 0))$ for any $\mathbf{a} \neq \mathbf{a}_\star$. This occurs once we've observed $(\mu, (\mathbf{a}, 0))$ for two distinct $\mu$ for each $\mathbf{a} \neq \mathbf{a}_\star$ and we've observed $(\mu, (\mathbf{a}_\star, \mu))$ for at least one $\mu$. We can compute an upperbound on the expected number of samples required to observe $(\mu, \tau_{\mathbf{a}}^{\text{exp}})$ for two distinct $\mu$ for each action sequence $\mathbf{a}$ by casting this as a coupon collector problem, where each pair $(\mu, \tau_{\mathbf{a}}^{\text{exp}})$ is a coupon. There are $2|\mathcal{A}|^H$ total coupons to collect. This yields that the expected number of samples is $\mathcal{O}(|\mathcal{A}|^H \log |\mathcal{A}|^H)$.

For $RL^2$, the exploration policy is optimal for all timesteps $t \geq T$ for some $T$ only if the exploitation values $\hat{V}_T^{\text{task}}(\tau^{\text{exp}} = (\mathbf{a}_\star, \mu)) = 1$ for all $\mu$ in $\mathcal{M}$. Otherwise, there is a small, but non-zero probability that $\hat{V}_t^{\text{task}}(\tau^{\text{exp}} = (\mathbf{a}, 0))$ will be greater at some $t > T$. For the exploitation values to be optimal at all optimal exploration trajectories $\hat{V}_T^{\text{task}}(\tau^{\text{exp}} = (\mathbf{a}_\star, \mu)) = 1$ for all $\mu \in \mathcal{M}$, we must jointly observe exploration trajectory $\tau^{\text{exp}} = (\mathbf{a}_\star, \mu)$ and corresponding action sequence $\mathbf{a}_\mu$ for each problem $\mu \in \mathcal{M}$. We can lower bound the expected number of samples required to observe this by casting this as a coupon collector problem, where each pair $(\tau^{\text{exp}} = (\mathbf{a}_\star, \mu), \mathbf{a}_\mu)$ is a coupon. There are

$|\mathcal{M}| \cdot |\mathcal{A}|^H$ unique coupons to collect and collecting any coupon only occurs with probability $\frac{1}{|\mathcal{A}|^H}$ in each episode. This yields that the expected number of samples is $\Omega(|\mathcal{A}|^{2H} \cdot |\mathcal{M}| \cdot \log |\mathcal{A}|^H)$. $\square$

## D EXTENDED RELATED WORK

**Detailed comparison with IMPORT and Humplik et al. (2019)** We compare DREAM with IMPORT (Kamienny et al., 2020) and Humplik et al. (2019) in greater detail, since they also leverage the problem ID. The key difference between DREAM and these works is that these works still learn exploration from returns achieved during exploitation, which leads to the chicken-and-egg problem. In contrast, DREAM learns exploration by maximizing the mutual information between exploration trajectories and learned $z$'s containing only task-relevant information (Section 4.2).

More specifically, like DREAM, IMPORT also conditions on the problem ID to learn exploitation without the need for exploration, which can accelerate learning exploration, but unlike DREAM, IMPORT still learns exploration from end-to-end signal, which can make learning exploration challenging as follows. Suppose that optimal exploitation conditioned on the problem ID is already learned, and exploration is learned from the exploitation returns. Since initially, the exploitation has not learned to interpret the trajectories produced by the exploration policies, it may still achieve low returns given a good exploration trajectory, which would erroneously "down weight" the good exploration trajectory (i.e., the chicken-and-egg problem). Humplik et al. (2019) attempts to learn better representations by predicting the problem ID as an auxiliary task. This can help learn a better policy, but does not address the chicken-and-egg coupling problem.

## E INFORMATION BOTTLENECK

The objective (Equation 2) for learning the exploitation policy and encoder is derived from minimizing the mutual information between problem IDs and the encoder's stochastic outputs, under the constraint that the exploitation policy achieves maximal returns, i.e.:

$$
\begin{aligned}
&\text{minimize} \quad I(z; \mu) \\
&\text{subject to} \quad \mathbb{E}_{\mu \sim p(\mu), z \sim F(z|\mu)} \left[ V^{\text{task}}(z; \mu) \right] \text{ is maximal.}
\end{aligned}
\tag{5}
$$

We form the Lagrangian with $\lambda^{-1}$ as the dual variable, which yields Equation 2. This constrained optimization problem is only satisfied with the Langrangian is maximized, as the dual variable tends to infinity: i.e., $\lambda$ approaches 0.

