# OpenReview forum: "Decoupling Exploration and Exploitation for Meta-Reinforcement Learning without Sacrifices"
_ICLR.cc/2021/Conference — Reject_

### Official Review · AnonReviewer2 · 2020-10-29
**Interesting and well-motivated paper with perhaps some missing exposition into non-meta exploration approaches**

**Rating:** 7
**Confidence:** 2

**Review:**

Summary: This paper introduces DREAM, a meta-RL approach that decouples exploration from exploitation. An exploitation policy learns to maximize rewards that are conditioned on an encoder that learns task relevant information. Then an exploration policy learns to collect data that maximizes the mutual information between the encoder and explored states. The work is compared against multiple baselines in simple tasks.


Overall, I lean towards accepting the paper, though I am not as familiar with the meta-RL literature to have much of an informed opinion about what relevant benchmarks or approaches are. The paper was well-written and well-motivated, and while the experiments were simple, seemed to highlight the problems that the paper was addressing. It makes sense to separate out exploration and exploitation and I appreciated the inclusion of tasks that helped motivate this point. Furthermore, the paper provides a theoretical analysis of DREAM showing that the decoupled policy maximizes returns. Code and hyperparameters are provided and the paper seems to be reproducible.


I do think that the paper should have more discussion and evaluation over approaches that aim to explicitly address the exploration exploitation problem. The paper only considers exploration in the context of meta-learning but of course exploration is a central problem in RL and several approaches have studied it outside of Meta-RL. The paper would be improved by discussing such approaches (for example intrinsic rewards such as empowerment [1] or surprise [2]) and/or evaluating how well these approaches compare to DREAM when trained alone and combined with vanilla algorithms.


I also would have liked to see more empirical analysis over the exploration policy being learned by $\pi^{exp}$.


Questions:


1) How was the decay rate for epsilon chosen in Figure 3? How would a policy with a fixed decay rate perform?


2) I do not quite understand how trajectories from the exploration policy can be used interchangeably with the encodings $z$ when plugged into $\pi^{task}$. Could the authors provide more insights into this?


[1] A Unified Bellman Optimality Principle Combining Reward Maximization and Empowerment. Leibfried et al.


[2] Curiosity-driven Exploration by Self-supervised Prediction. Pathak et al.

---

> ### Author Response · Authors · 2020-11-15
> **Author Response**
>
> We thank R2 for their review. R2 finds this work “well-written,” “well-motivated,” and “reproducible.” We believe that we have addressed all concerns below and have updated our draft accordingly (edits in blue for convenience). We ask R2 to please let us know if there are any remaining concerns.
>
> **Comparison to other exploration works outside of meta-RL.** We have included a discussion of other exploration works outside of meta-RL in our updated draft in the related work (Section 2), which we summarize below. Just like exploration in normal RL, the goal of exploration in meta-RL is to gather informative data (the exploration trajectory) that enables learning a good policy (the exploitation policy, conditioned on the exploration trajectory). The key difference with prior work (Liebfried et al., 2019; Pathak et al., 2017; Bellemare et al., 2016; Burda et al., 2018) on exploration outside of meta-RL is that exploration in meta-RL leverages prior experience from meta-training to perform more targeted exploration. For example, in a new kitchen, a robot chef needs not explore the entire kitchen, and may instead only explore the cabinets and pantry to find the ingredients. In contrast, in normal RL, the agent starts tabula rasa, so prior works on exploration broadly seek to visit many new states. Therefore, DREAM can more quickly adapt to a new task (e.g., kitchen) from leveraging its prior experience than prior works on normal RL exploration. In the experiments, by leveraging experience from meta-training, DREAM learns to solve a new MDP in **just two episodes**, while tabula rasa RL approaches require many more episodes. Therefore, due to the differences in setting, a direct empirical comparison between DREAM and tabula rasa RL exploration approaches does not seem informative.
>
> **Analysis of the learned exploration policy.** Appendix B.2 includes qualitative analysis of the learned exploration policy. Additionally, the website linked in the paper includes videos (https://anonymouspapersubmission.github.io/dream/) that compare the learned policies of DREAM with other approaches.
>
> **Questions:**
> - **“How was the decay rate for epsilon chosen in Figure 3?”** We chose a fixed epsilon of 0.1. During the rebuttal, we also tried varying the fixed value of epsilon from 0.1 to 1 in increments of 0.1. As epsilon increases, the gap slightly narrows between DREAM and end-to-end approaches, since DREAM quickly learns optimal exploration, so following it on-policy helps learn exploitation more quickly. However, as the theory predicts, DREAM still learns optimal exploration and exploitation in $\Omega(|\mathcal{A}|^H)$ fewer samples than RL^2, regardless of the value of epsilon. **“How would a policy with a fixed decay rate perform?”** We also experimented with an epsilon that decays from 1 to 0.1 in a fixed number of episodes, which we varied between 1K, 10K, 100K. Over all such decay rates, DREAM still learns more quickly by $\Omega(|\mathcal{A}|^H)$ fewer samples.
> - **How can the exploration trajectories be swapped for the encodings $z$ during meta-test time?** Recall that the encodings $z$ are learned to encode only the task-relevant information, and then the exploration policy is trained to produce exploration trajectories that have high mutual information with z (i.e., recover the task-relevant information). Specifically, DREAM also learns a decoder $q(z \mid \tau^\text{exp})$. When this decoder is parametrized as a Gaussian centered around a learned embedding, optimizing the DREAM objective produces the same encodings z from the exploration trajectory: i.e., the L2 distance between the decoder’s $z$’s and encoder’s $z$’s is minimized (see item 2 in Section 4.3). Hence, $z$’s sampled from the decoder (computed from the exploration trajectory) can be directly swapped with the encoder’s $z$’s during meta-test time.

---

> ### Author Response · Authors · 2020-11-19
> **Request for Discussion**
>
> We believe that we’ve addressed all of your concerns in the response below and revised paper draft. Please let us know if there are any remaining concerns or questions! We would especially appreciate a response before the discussion period ends on Tuesday so that we can clarify any further questions.

---

### Official Review · AnonReviewer4 · 2020-10-29

**Rating:** 6
**Confidence:** 3

**Review:**

The paper investigates the exploration-exploitation problem in meta-learning. The authors explain the problem of coupled exploration and validate it through a toy example. To overcome this issue, the paper introduces DREAM, a meta-algorithm decoupling exploration and exploitation. In the first step, DREAM learns an exploitation policy and a task embedding by maximizing the cumulative reward of the given task (task identifier is known at train). In the second step, DREAM learns an exploration policy that is "compatible" with the embeddings generated by the exploitation policy. DREAM outperformed the state-of-the-art algorithms in several experiments.

The paper is well written and easy to follow. The idea is clearly explained and justified. I have only a few comments.

Concerning the problem of coupled exploration, I'm wondering if you could provide a more formal justification. The current justification (sec 4.1) is easy to follow but quite abstract. Despite being supported by your example (Sec 5.2), it is not clear to me whether this is a general problem of coupled exploration. Eg, is it due to gradient updates?

Comparison with the literature: could you provide a detailed comparison with (Humplik et al., 2019; Kamienny et al., 2020)?
In the first step, your algorithm learns an encoding of the task $f_\psi(\mu)$ by maximizing the task reward (ie learning the exploitation policy). In the second step, it uses this embedding to train an exploration policy to generate trajectories mapping to the relevant information constructed by the embedding (ie maximizing the mutual information between $z$ and the trajectories). To the best of my understanding, this seems very similar to what done in the mentioned approaches. In particular, (Kamienny et al., 2020) also have exploration and exploitation policies. The exploitation policy is trained to maximize the reward while generating an embedding of $\mu$, thus similarly to your exploitation step. The exploration policy is also trained to maximize the reward while enforcing the RNN state to be similar to $f(\mu)$. More mildly, even their approach aims to learn to generate trajectories providing information about $z$. The main difference between the two approaches resides in the fact that their exploration policy is trained to also maximize the reward, am I correct? Will their algorithms suffer from the coupling problem mentioned in section 4.1?


Minor comments
Proposition 1: could you explain in more details the need for ergodicity? My understanding from the appendix is that you should be able to generate all possible trajectories to have that $p(z|\tau^{exp}) = F_\star(z|\mu)$. However, you would need to potentially enforce exploration at the level of actions. This can be obtained by a random policy, am I correct?

I think $\mathbb{E}_{\mu \sim p(\mu)}$ is missing in the first and second equations of the proof of Proposition 1.

You mentioned that you could remove the "ergodicity assumption by increasing the number of exploration episodes". Could you clarify this sentence? The number of exploration episodes is a term that does not appear in your current analysis since the reasoning seems to be "in expectation".

I didn't check the Appendix C.2

---

> ### Author Response · Authors · 2020-11-15
> **Author Response**
>
> We thank R4 for their review. R4 finds that this work is “clearly explained and justified,” and leaves several comments, which we address below. We have also updated the paper accordingly (edits in blue for convenience). We ask R4 to please let us know if there are any remaining concerns.
>
> **R4 asks for a formal justification for the chicken-and-egg problem.** The high-level intuition is that end-to-end approaches learn exploration from a quantity that is challenging to estimate (the exploitation returns), while DREAM learns the exploration policy from a quantity that is much easier to estimate (i.e., the decoder). Section 5.3 formally illustrates this with the tabular Q-learning example. In end-to-end approaches, the exploration Q-value regresses toward the exploitation Q-value, which requires many samples to correctly estimate. As a result, the exploration Q-value regresses toward the wrong value for many timesteps. In contrast, for DREAM, the exploration Q-value regresses toward the decoder, which, in this example, requires only a single sample to accurately estimate. Consequently, the DREAM exploration Q-value regresses toward the correct value immediately. Overall, this effect holds whenever exploration is learned directly from the exploitation returns (all end-to-end approaches), whereas DREAM avoids this by learning exploration from an intermediate reward based on the decoder. We welcome any feedback on making this chicken-and-egg problem clearer in the text.
>
> **R4 asks for a detailed conceptual comparison with Humplik et al., 2019 and Kamienny et al., 2020.** R4 is correct: the key difference from DREAM is that these approaches still learn exploration based on the exploitation rewards, so they still suffer from the chicken-and-egg coupling problem. Instead, DREAM learns exploration by maximizing the mutual information between exploration trajectories and learned z’s containing only task-relevant information. We have included an extended related works section in Appendix D to clarify this.
>
> **R4 asks why Proposition 1 requires assuming that the problems are ergodic and for clarification about the relationship between the ergodicity assumption and the number of exploration episodes.** In the worst case, inferring z from the exploration trajectory with the decoder may require visiting each transition many times, e.g., if the dynamics are highly stochastic. Therefore, it must be possible to revisit every state from every other state (i.e., ergodicity, defined by Moldovan et al., 2012). Alternatively, these transitions can be visited many times from the initial state(s) if there are many exploration episodes. In the proof of Proposition 1, this work assumes the former: that the problems are ergodic and there is only one exploration episode (from Section 3).
>
> We thank R4 for pointing out the errata with the proof of Proposition 1, which we have fixed in the updated draft.

---

> > ### Comment · AnonReviewer4 · 2020-11-20
> > **Response**
> >
> > Thank you for the clarifications.
> >
> > I have another question about the "chicken-and-egg problem". You added the following sentence to the revised version of the paper referring to DREAM: "There is no chicken-and-egg effect in this joint training because the exploitation policy (along with the stochastic encoder) are trained independently from the exploration policy; the training of the exploration policy only uses the stochastic encodings." The exploration policy depends on the exploitation policy through the encoder $F_{\psi}(z | \mu)$ so there is still a dependence. However, if I understand correctly, the chicken-and-egg effect works only in one direction and thus this dependence is not problematic. Could you please explain a little bit better this point?
> >
> > Figure 6 reports the average return as a function of time. Could you clarify how it is computed? How many trials are included? Does it include both exploration and exploitation trials?

---

> > > ### Author Response · Authors · 2020-11-21
> > > **Further clarifications**
> > >
> > > We thank R4 for their response.
> > >
> > > **Clarifying the chicken-and-egg problem.** To clarify, the chicken-and-egg problem occurs in end-to-end approaches since (i) exploitation can only be learned efficiently when exploitation is already good and (ii) the learning signal for exploration (i.e., the expected exploitation returns) is challenging to estimate. DREAM clearly avoids (i) by learning the exploitation policy (and encoder) independently from exploration, and hence they quickly converge to their optimal values. To avoid (ii), once the encoder is learned, it forms a learning signal for exploration separate from the exploitation returns; Section 5.2 illustrates how this allows the exploration policy to learn in far fewer samples. As R4 notes, the learning signal for exploration may not be perfect until the encoder has learned, but again, this occurs quickly, since it is learned independently from exploration. In practice, rather than waiting for the exploitation policy and encoder to converge before training the exploration policy, we train the exploration policy even using the sub-optimal encoders from early iterations for convenience. To re-emphasize, this training of the exploration policy does not affect or slow down the learning of the encoder/exploitation policy.
> > >
> > > **Clarifying the training curves in Figure 6.** The y-axis for the middle and right of Figure 6 is the average returns achieved over 100 meta-testing trials, which we periodically evaluate every 2000 meta-training trials. The y-axis for the left of Figure 6 is the performance of each approach on the meta-training problems, which is logged every 10 meta-training trials. The x-axis is the number of total timesteps spent in all meta-training trials up to this point, including timesteps from both exploration and exploitation. We have updated the draft to include this information.

---

> ### Author Response · Authors · 2020-11-19
> **Request for Discussion**
>
> We believe that we’ve addressed all of your concerns in the response below and revised paper draft. Please let us know if there are any remaining concerns or questions! We would especially appreciate a response before the discussion period ends on Tuesday so that we can clarify any further questions.

---

### Official Review · AnonReviewer3 · 2020-10-29
**Well-written paper. Attempts to address an important problem in meta-learning. Has room for improvement to produce a stronger version.**

**Rating:** 4
**Confidence:** 4

**Review:**

The paper introduces an approach to improve meta-learning in RL. Specifically, the approach aims to improve the agent’s exploration during the training phase, so that the agent can better exploit during the test phase where samples collected from the training phase provides useful task-relevant information to the agent.

Pros:
The paper is well-written.

The idea is clearly presented.

The experiments are clear to understand.

The results presented demonstrate that the approach can either match or improve learning when compared to relevant baselines.

Cons:
Details about the architecture choices used for the baselines are not provided in the main text, which makes it difficult to understand the experiment setup and its results.

Drawing out the differences between the baselines and their approach would be useful to help understand the contribution of the work.

Questions:
1. It seems like the use of problem ID is specific to the approach introduced in the paper. If so, could the authors describe how this was used in the baseline agents? If the problem IDs were not used in the baselines, then it feels like the baselines considered here are unfair.
2. For the domains considered here, it seems like the problem IDs are sufficient enough to provide any task-relevant information to the meta-learning agent. In this case, what sort of z’s can the encoder produce? The domains considered seem less interesting for the introduced approach, as it seems like the encoder could simply learn an identity mapping of the problem ID (which is its input).
3. What would be the reason for RL^2 to fail in the 3D navigation task considered?
4. As a baseline, it would be interesting to see the learning performance of a simple RL^2 agent on the domains, provided they take in the problem ID as input? This would inform whether the encoder-decoder architecture that is introduced in the paper learns something that is beyond the problem ID.

---

> ### Author Response · Authors · 2020-11-15
> **Author Response**
>
> We thank R3 for their review. R3 finds this work “clearly presented” and that “[DREAM] can either match or improve learning,” though they raise some concerns and questions. We believe that we have addressed all concerns below, and we have updated the draft accordingly (edits in blue for convenience). We ask R3 to please let us know if this is the case, or if there are any additional questions / concerns.
>
> **Architecture details.** All of the methods (DREAM and the state-of-the-art methods compared against) use the same architecture where applicable. We have updated the main text to make this clear and also refer the reader to the full details in Appendix B.3, when introducing the other approaches (Section 6).
>
> **How does DREAM differ from prior approaches?** As described in Section 2, prior approaches fall into two main categories: (i) *end-to-end approaches*, such as RL^2, VariBAD, IMPORT, which can represent the optimal policy, but require many samples to escape local optima due to the chicken-and-egg problem; and (ii) *decoupled but suboptimal approaches*, such as PEARL (Rakelly et al., 2019), which are more sample efficient from avoiding the chicken-and-egg problem, but do not learn the optimal policy, even with infinite meta-training data. In contrast, the main contribution of this work is that DREAM achieves the best aspects of each: it is **sample efficient from avoiding the chicken-and-egg problem, and also optimizes an objective that can learn the optimal exploration and exploitation strategies with sufficient meta-training data**. We are happy to revise the related work (Section 2), if R3 finds this unclear. We have also included an extended related works section, with detailed differences between IMPORT, Humplik et al., 2019, and DREAM in Appendix D.
>
> **Questions:**
> 1. **How is the problem ID used by approaches we compare against?** To ensure a fair comparison, this work compares against two approaches that use the problem ID: IMPORT (Kamienny et al., 2020) and an upper bound on PEARL (Rakelly et al., 2018). IMPORT learns a policy conditioned on the problem ID during meta-training, like the DREAM exploitation policy. However, IMPORT still learns exploration end-to-end, so it suffers from the chicken-and-egg problem. PEARL maintains a separate replay buffer for each separate problem ID in order to form many different contexts for its encoder during meta-training. Furthermore, access to the problem ID is realistic in real-world meta-RL tasks. For example, a recommendation system may want to quickly adapt to new users (tasks), which are uniquely identifiable (e.g., by their usernames).
> 2. **What sorts of z’s does the encoder produce — is learning an identity mapping of the problem ID sufficient?** Recall that the problem ID indirectly contains all task-relevant information, but also potentially task-irrelevant information, which we try to remove with the information bottleneck and exploitation objective. For example, in the distracting bus benchmark, different problems (and hence different problem IDs) differ in the locations of both the helpful (colored) buses, and the unhelpful (gray) buses, which are never used to optimally solve any problem. The DREAM encoder learns to output z’s that only encode the locations of the helpful buses. Here, an identity mapping is not sufficient, since this would encourage the exploration policy to explore the unhelpful gray buses, which leads to low returns, since the exploration episode is not long enough to explore all buses, both helpful and helpful. Removing the information bottleneck, which allows learning an identity mapping, empirically substantially hurts performance (See DREAM (no bottleneck) in Figure 5). Figure 11 in the Appendix also illustrates the z’s that the encoder (with the information bottleneck) learns. They are clustered into 4! clusters, corresponding to the different helpful bus permutations.
> 3. **Why does RL^2 fail in the 3D navigation task?** We revised the experiments section to clarify why RL^2 fails. Specifically, as discussed in Section 4.1, end-to-end approaches, including RL^2, become stuck in a local optimum due to the chicken-and-egg problem. At the beginning of training, they struggle to learn to explore and walk around the barrier to read the sign, since they do not know how to use this information to solve the task (exploitation), and consequently the reward signal for exploration is misleading. On the other hand, since they haven't learned to walk around the barrier to read the sign, they cannot learn to use this information. In other words, both exploration and exploitation are challenging to learn without already having learned the other.
> 4. **What about a baseline that uses the problem ID during meta-testing?** A baseline that uses the problem ID during meta-testing could not generalize to new held-out problems, since the held-out problem ID is a novel one-hot ID (e.g., in the cooking and distracting bus experiments).

---

> ### Author Response · Authors · 2020-11-19
> **Request for Discussion**
>
> We believe that we’ve addressed all of your concerns in the response below and revised paper draft. Please let us know if there are any remaining concerns or questions! We would especially appreciate a response before the discussion period ends on Tuesday so that we can clarify any further questions.

---

### Official Review · AnonReviewer1 · 2020-11-03
**This paper contains interesting results on exploration of meta-RL, but the presentation needs to be fixed**

**Rating:** 5
**Confidence:** 4

**Review:**

Summary of the authors claim\
The authors claim end-to-end learning of exploration and exploitation policies can result in a poor local minima. To avoid this problem, the authors propose to separate the objectives for exploration and exploitation in meta-RL. Theoretical analyses and the experiments on several artificial tasks show the effectiveness of the proposed method and superiority over existing methods.

Overview of the proposed method\
As is common in the literature, the exploitation policy depends not only on observations, but also on a task-embedding $z$ to adapt the environment across the tasks. $z$ is generated from a stochastic encoder which is conditioned on the problem ID $\mu$. By minimizing the mutual information between the problem ID $\mu$ and the task-embedding $z$ which is called the information bottleneck term, the exploitation policy tends to be independent from the task irrelevant information and can reduce the risk of being trapped at a poor local minima. On one hand, the lower bound of the mutual information between the exploration experiences $\tau^{exp}$ and the task-embedding $z$ is maximized for the learning of the exploitation policy. During this optimization, decoder q(z|\tau^{exp}) is trained and this decoder is used to estimate the task-embedding without knowing the problem ID. Because the exploitation policy depends on the task embedding that is generated from problem ID and not on the exploration experiences $\tau^{exp}$, it make the training of the exploitation policy more robust and  successful.

Comments on the paper\
I appreciate the good experimental results.\
But I think the paper contains inaccurate explanations and the proposed method is not properly characterized.
- Although the authors emphasize that the decoupling of the exploration and exploitation is important in Section 1, Section 4.1 and also in the title, the actual proposed algorithm has coupled loss functions.
In Section 4.1, the authors wrote “To solve the task, $\pi^{task}$ needs good exploration data from a good exploration policy $\pi^{exp}$.” Rigorously speaking, this is not true. Even if the exploration policy is just uniform random distribution, we know table Q-learning algorithms can solve the task when we have an infinite number of samples. If the statement means “To **efficiently** solve the task, $\pi^{task}$ needs good exploration data from a good exploration policy $\pi^{exp}$.” It will be true, but this is also true for the proposed algorithm when $z$ is computed by not only using the encoder $F(z|\mu)$, but also by using $g(\tau^{exp})$. This makes the training of $\pi^{task}$ more coupled training.  I wonder how much this mixed training heuristics is important. It would be clear if the authors provide the experimental results with and without this heuristics.
Another statement, “Learning $\pi^{exp}$ relies on gradients passed through $\pi^{task}$.” is true. If the embedding $z$ is not informative for solving the task, that is, $\pi^{task}$ does not make use of $z$, it is not meaningful to maximize the (lower bound of) mutual information between $z$ and the experience by the exploration policy $\pi^{exp}$, $\tau^{exp}$.  Thus it is also true for the proposed method. Therefore, it seems for me the statement not only explains the reason why the existing methods sometimes does not work, but also explain the reason why the reinforcement learning is difficult. I think the stress should be put on the training by using the problem ID which does not depend on the progress of the training rather than the decoupling.
- The condition of Proposition 1 is not clearly stated. In the proof, the authors assume “$\lambda$ approaches 0”, but this is not clearly stated in the condition. Also in the experiment, this is not performed, but $\lambda$ is set to be 1.
- Usefulness of the bottleneck (as shown in Fig.5) is not supported by the theoretical analyses. It seems for me theoretical analysis does not support the actual proposed method which uses the bottleneck in the optimization. Instead, as I wrote above, the authors emphasize the importance of the decoupling too much. However, based on the theoretical analysis and the experimental results, I think the authors should emphasize more the usefulness of the bottleneck.
- To use the proposed algorithm, Problem ID is indispensable. This condition should be stressed. I also wonder how much the performance of the proposed method is affected by the mapping of the problem ID. Is it robust to the random permutation of the problem ID? It would be nice if the authors can test the robustness on the permutation of the problem ID.
- As written in the introduction, the meta-reinforcement learning is often inspired by the humans' quick adaptation ability to new task which shares some properties with the previously experienced tasks. However, the necessity of the problem ID limits the applicability of the method to such situations. It would be nice if the authors can exemplify what kind of practical tasks the proposed method can be applied.
- As for the experiments, it is good to test on newly designed environments and tasks to highlight the properties of the proposed method. However, I believe the proposed method should be tested on one of the common benchmarks to make the experimental results more reliable. From these experiments, the readers will understand that the existing methods are implemented appropriately and how much the proposed method works well on the common benchmark.

---

> ### Author Response · Authors · 2020-11-15
> **Author Response (Part 1 of 2)**
>
> We thank R1 for the detailed review. We have addressed all of R1’s concerns below, and have also updated the paper accordingly (edits are written in blue for convenience). We ask R1 to please let us know if there are any remaining concerns.
>
> R1 is concerned that **exploitation and exploitation may still be coupled in DREAM** for two reasons, which we address below and in the updated draft. We emphasize that DREAM’s form of decoupling does avoid the chicken-and-egg coupling problem.
> - (i) DREAM consists of two stages (Section 4.2). First, the encoder and exploitation policy are jointly trained until the exploitation policy has converged and the encoder learns to extract task-relevant information. Second, the exploration policy is trained to recover the task-relevant information from the encoder. R1 correctly notes that successfully training the exploration policy in the second stage does require an informative encoder from the first stage. **However, training the exploration policy to maximize the mutual information with the encoder, rather than from exploitation returns (as end-to-end approaches do) is critical for avoiding the chicken-and-egg problem.** Suppose that an optimal exploitation policy is learned in the first stage, but the exploration policy learns based on the exploitation returns instead of DREAM’s second stage. Since initially, the exploitation policy has not learned to interpret the trajectories produced by the exploration policy, it may still achieve low returns given a good exploration trajectory, which would erroneously “down weight” this good exploration trajectory (i.e., the chicken-and-egg problem). We note that this work’s implementation of DREAM runs the two stages of DREAM simultaneously, since this achieves good performance and is simpler, but the two stages are conceptually sequential.
> - (ii) As a heuristic to aid training, DREAM alternates between training the exploitation policy conditioned on z sampled from the encoder and decoder. We have removed this heuristic and found that **DREAM’s performance actually improves** when the exploitation policy only trains conditioned on the z’s from the encoder’s output, which further suggests that decoupling is beneficial. We have updated all experiments to use DREAM without this heuristic.
>
> **How realistic is knowing the problem ID during meta-training?** We emphasize that the problem IDs are just for distinguishing between different problems, implemented as simple unique one-hots: i.e., they are the integers 1, 2, …, # problems. In real-world meta-RL applications, such a problem ID is typically easily available. For example, we may want a recommendation system that can quickly adapt to new users (tasks). These users are typically uniquely identifiable by their username, which could serve as a problem ID. For a robot chef, we may want to train the robot in many different kitchens (tasks) in a factory and each kitchen could easily be assigned a unique ID.
>
> Additionally, we emphasize that other works also use this problem ID — which this work compares against in the experiments — such as PEARL (Rakelly et al., 2018), IMPORT (Kamienny et al., 2020), and Humplik et al., 2019. Therefore, DREAM’s improved performance is not simply from additional information, but rather, from effectively using this information.
>
> Finally, R1 suggests **more heavily stressing that DREAM uses problem ID.** We are happy to stress this where R1 feels appropriate. We note that the current work already emphasizes that DREAM uses the problem ID: e.g., when first introducing DREAM in the introduction, in the description of the setting (Section 3), in Section 4.2 detailing DREAM, and in the conclusion.
>
> **R1 asks if DREAM is robust to variations in the problem ID.** Specifically, R1 asks if DREAM is robust to permutations of the problem ID. The problem IDs are one-hot IDs with arbitrary ordering, so DREAM is robust to permutations, since it does not rely on ordering in any way. During the discussion period, we also tested a second form of varying the problem ID, by assigning each problem in the map benchmark 3 different problem IDs. When a problem is sampled during meta-training, it is randomly assigned 1 of those 3 different IDs. DREAM still achieves optimal exploitation returns and outperforms prior state-of-the-art approaches (IMPORT, VariBAD, E-RL^2, PEARL) by learning to map each of those 3 IDs to the same z in the encoder. We have added this experiment to the updated draft in Appendix B.2.

---

> > ### Author Response · Authors · 2020-11-15
> > **Author Response (Part 2 of 2)**
> >
> > **R1 suggests greater emphasizing the information bottleneck.** Both the information bottleneck and the decoupling aspect are critical to DREAM, though this work focuses on the decoupling aspect, since this addresses the failure mode of prior approaches. This work analyzes them separately, to better understand the importance of each. The theoretical analysis (Section 5.2) formally shows the sample complexity effect of decoupling and how it scales with the action space, and horizon, assuming a perfect information bottleneck. In practice, learning such a perfect information bottleneck may be challenging, so this work empirically evaluates this on the distracting bus benchmark by comparing DREAM with and without the information bottleneck (Figure 5).
> >
> > **Existing benchmarks.** To ensure that all methods are correctly implemented, we validate them on benchmarks from prior papers:
> > - The 3D visual navigation benchmark is derived from Kamienny et al., 2020. The results in this work on this benchmark are consistent with those reported in Kamienny et al., 2020: in the hardest version of their task, they also find that IMPORT and RL^2 cannot solve the task.
> > - We evaluated this work’s implementation of VariBAD on the grid world benchmark from the VariBAD paper (Zintgraf et al., 2019) and found comparable performance.
> > - We evaluated this work’s implementation of E-RL^2 on the multi-arm bandit benchmark from the RL^2 paper (Duan et al., 2016) and found comparable performance.
> >
> > Since all approaches are implemented for discrete action spaces, we do not validate them on continuous control benchmarks, but we are happy to validate them on any other open source discrete control benchmarks. We note that this work evaluates on new benchmarks designed to stress test exploration in meta-RL (the focus of this work), since prior meta-RL works are not designed to test exploration (details in Section 6).
> >
> > **“The condition of Proposition 1 is not clearly stated”:** We thank R1 for pointing this out, which we clarify below and in the updated draft. The information bottleneck is derived from a constrained optimization problem, which we optimize by forming the Lagrangian, with dual variable $\lambda^{-1}$. The Lagrangian must be minimized as $\lambda^{-1}$ approaches infinity ($\lambda$ approaches 0) to solve this optimization problem, which is used in the proof of Proposition 1. In the experiments $\lambda$ is set to 1 for simplicity, since this achieves good performance, but it could be adjusted with dual gradient descent.
> >
> > We have additionally updated the draft with the phrasing suggestions from R1.

---

> ### Author Response · Authors · 2020-11-19
> **Request for Discussion**
>
> We believe that we’ve addressed all of your concerns in the response below and revised paper draft. Please let us know if there are any remaining concerns or questions! We would especially appreciate a response before the discussion period ends on Tuesday so that we can clarify any further questions.

---

### Author Response · Authors · 2020-11-17
**Updated Draft Addressing Reviewer Concerns Posted**

We thank all the reviewers for their time. Overall, the reviewers found this work “__interesting__,” “**well-written** and **well-motivated**,” and that the “theoretical analyses and the experiments … show the … **superiority [of the proposed approach] over existing methods**,” and also raised several concerns / questions.

We addressed all concerns / questions in individual review responses and updated the draft accordingly (see edits in **blue**). We look forward to discussion with the reviewers. Below, we summarize the main changes (see details in individual responses).

- We clarified a **misunderstanding about whether DREAM is decoupled (R1)**. We clarified that **DREAM is indeed decoupled**. Although DREAM is conceptually decoupled, the experiments originally used a heuristic that slightly couples exploration and exploitation in DREAM. We completely removed this heuristic in the updated draft, and found that **DREAM’s performance further improves**.

- We **clarified the chicken-and-egg problem (R4)**.

- We addressed concerns about **DREAM’s use of the problem ID (R1 / R3)**. We clarified that access to **the problem ID (a simple unique one-hot) can be realistic in real-world meta-RL problems**. Furthermore, the experiments show that **DREAM outperforms other prior works that also use the problem ID**.

- Finally, we **extended the related works** to compare with **IMPORT and Humplik et al., 2019 in greater detail (R3 / R4)** and to discuss work on **exploration outside meta-RL (R2)**.

---

### Decision · Program_Chairs · 2021-01-07
**Final Decision**

**Decision:**

Reject

**Comment:**

The authors clarified many of R1 and R4's concerns, but there were important remaining concerns regarding the presentation.

On the bright side, the approach is novel and the experimental results are solid.

However, the main point raised by R1 is the mismatch between the narrative and the theory and the actual algorithm and results. Some exemples of this mismatch include:
- Proposition 1 is proved when lambda goes to 0, which is never mentioned in the main paper. One has to look into the appendices to have a discussion of lambda and of the algorithm, while the authors could clearly explain that when discussing the theoretical result. The added discussion on tuning of lambda based on Appendix E does not help because the optimization problem is written as "under the constraint that [...] is maximized", which is not particularly clear.
- More generally, the theoretical result (including e.g., the assumption of ergodicity) could be discussed more precisely in terms of what is actually done in the algorithm and the experiments.
- as stated by R3, many important aspects of the methods and the experimental setup are only available in appendices, which makes it difficult to understand the similarities and differences in experimental protocol between the curent paper and RL^2, PEARL and IMPORT.
- it is unclear what part of DREAM is critical for performance. There is no thorough ablation study nor discussion of the importance of the information bottleneck term, and the only signal given in Figure 5 is that it is critical. The authors could clarify the two aspects (decoupling and information bottleneck).


There was some discussion about this paper, but even under the assumption that the authors answered most R3's concerns (R3 didn't engage in discussions), the paper is still borderline. In the end there was little support for acceptance because of the presentation issues above.